# Local and global reward learning in the lateral frontal cortex show differential development during human adolescence

Marco K. Wittmann[1,2,3,4☉], Maximilian Scheuplein[1,5☉], Sophie G. Gibbons[1,6], MaryAnn P. Noonan[1,7] *

1 Department of Experimental Psychology, University of Oxford, Radcliffe Observatory, Oxford, United Kingdom, 2 Wellcome Centre for Integrative Neuroimaging, University of Oxford, John Radcliffe Hospital, Headington, Oxford, United Kingdom, 3 Department of Experimental Psychology, University College London, London, United Kingdom, 4 Max Planck UCL Centre for Computational Psychiatry and Ageing Research, University College London, United Kingdom, 5 Institute of Education and Child Studies, Leiden University, Leiden, the Netherlands, 6 MRC Cognition and Brain Sciences Unit, University of Cambridge, Cambridge, United Kingdom, 7 Department of Psychology, University of York, York, United Kingdom

☉ These authors contributed equally to this work.
* maryann.noonan@psy.ox.ac.uk

**Data Availability Statement:** Summary data are available as a supplemental excel spreadsheet (S1 Data) in which each tab summarises the data to

## Abstract

Reward-guided choice is fundamental for adaptive behaviour and depends on several component processes supported by prefrontal cortex. Here, across three studies, we show that two such component processes, linking reward to specific choices and estimating the global reward state, develop during human adolescence and are linked to the lateral portions of the prefrontal cortex. These processes reflect the assignment of rewards contingently to local choices, or noncontingently, to choices that make up the global reward history. Using matched experimental tasks and analysis platforms, we show the influence of both mechanisms increase during adolescence (study 1) and that lesions to lateral frontal cortex (that included and/or disconnected both orbitofrontal and insula cortex) in human adult patients (study 2) and macaque monkeys (study 3) impair both local and global reward learning. Developmental effects were distinguishable from the influence of a decision bias on choice behaviour, known to depend on medial prefrontal cortex. Differences in local and global assignments of reward to choices across adolescence, in the context of delayed grey matter maturation of the lateral orbitofrontal and anterior insula cortex, may underlie changes in adaptive behaviour.

## Introduction

A distributed network in the human brain supports learning from reward and making adaptive decisions. This network comprises several regions in lateral and medial prefrontal cortex (PFC), including lateral and medial orbitofrontal/ventromedial prefrontal cortex, as well as other areas such as anterior cingulate cortex, insula cortex, and the amygdala. In concert, they contribute component parts to adaptive behaviour such as contingency learning, value comparison, and value representations [1–8]. However, so far, we only have rudimentary

reproduce each figure. Multiple figure panels are included on each tab.

**Funding:** This work was supported by the Academy of Medical Sciences (AMS_SBF003_1143 to MPN), the John Fell Fund from the University of Oxford (0011041 to MPN) and the Wellcome Trust (WT100973AIA to MKW). The funders had no role in study design, data collection and analysis, decision to publish, or preparation of the manuscript.

**Competing interests:** The authors have declared that no competing interests exist.

**Abbreviations:** GLM, general linear model; GRS, global reward state; LME, linear mixed effects; PFC, prefrontal cortex; ROI, region of interest.

knowledge about the developmental dynamics of this brain network and accompanying behavioural changes during adolescence and early adulthood [9,10].

Here, we focus on the development of component processes of reward learning that have been strongly linked to neighbouring regions of orbitofrontal and anterior insula cortex in studies of nonhuman primates: local and global reward learning. Local reward learning refers to the ability to form contingencies between choice options and outcomes, repeating choices that led to positive outcomes and omitting choices that led to negative outcomes [11–13] (also referred to as "contingent reward learning" or "contingent credit assignment"). By contrast, global reward learning refers to a parallel mechanism where reward simultaneously reinforces not only the choice that caused it but also unrelated choices made in close temporal proximity [6,7,14–17]. This noncontingent global reward learning involves forming a representation of the global reward state (GRS), i.e., how much reward was received overall recently independent of the specific choices that caused them [7]. Lesion studies in macaques and human patients have consistently causally linked local reward learning to lateral orbitofrontal cortex [6,7,14,17], and this is even engrained in variations of grey matter volume in these regions [18]. By contrast, global reward learning mechanisms have been associated with BOLD activity in neighbouring anterior insula cortex [7]. Notably, the function of both these regions contrasts with medial orbitofrontal/ventromedial prefrontal cortex, which harbours a variety of value signals linked to value comparison and decision-making processes, as opposed to learning processes [14,16,19–25].

Informed by these animal models and the precise functional localisation of these mechanisms, we consider the development of reward-guided learning in the context of the protracted [26–33] and nonuniform [31,33] structural maturation of the brain. These considerations lead to the hypothesis that specific cognitive abilities, particularly those related to lateral prefrontal cortex, mature later than others, in particular, the more medial regions [31,33,34]. This temporal mismatch between protracted structural changes in prefrontal cortex and more rapid maturation of subcortical areas has been suggested to account for increased risk-taking behaviour in adolescence [35,36], and several studies link development of reward-related behaviour to changes in prefrontal–subcortical interactions [37–39]. However, prefrontal cortex has often been treated as a unitary structure, and, consequently, we only have a coarse understanding of the different speeds at which subregions of the frontal cortex and, in parallel, subcomponents of reward learning mature [40]. With tentative evidence that adolescents differ from adults in terms of local reward learning, for instance, in terms of balancing positive and negative feedback [41–43,43–46], it becomes critical to understand the development of reward learning mechanisms in combination with developmental maturation of their neural underpinnings.

Here, we combined behavioural and lesion investigations to suggest an important role for neighbouring subregions of the lateral frontal cortex, specifically orbitofrontal and anterior insula cortex in the development of local and global reward learning. We used the same multi-option probabilistic learning task originally developed in macaques to dissociate local and global reward learning (Fig 1) [4,6,14]. In study 1, we tested a large online sample (overall $n = 422$) of adolescents (11 to 17 years) and young adults (age 18 to 35 years) and showed that both local and global reward learning change during human adolescence. We chose this age range in accord with previous work [47] and with particular reference to the protracted maturation profile of lateral prefrontal cortex [30]. Our findings suggest that young adults associated choices more strongly with local rewards and were simultaneously negatively influenced by the GRS. The GRS influence became even more negative with age, meaning young adults, more than adolescents, contextualised their choices within the longer-term reward context and were less likely to persist with a choice if the alternatives afforded by this context were attractive. By contrast, decision bias mechanisms that depend on the ventromedial portions of the orbitofrontal cortex showed no relationship with age.

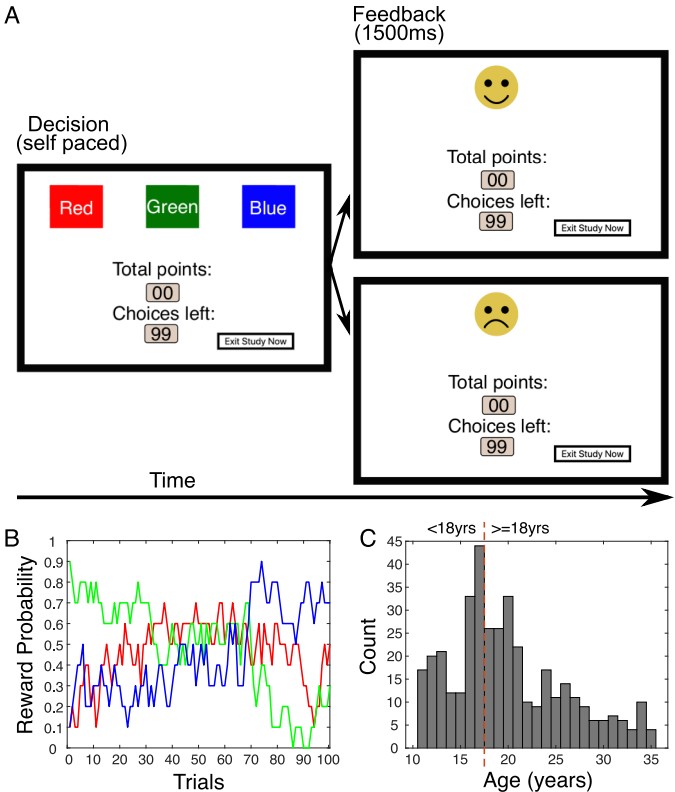

**Fig 1. Task design, reward schedule, and sample. (A)** Trial timeline: Participants decided between three choice options (red, green, blue squares; left-hand side) before receiving feedback for 1,500 ms (right-hand side) indicating whether their choice yielded a reward (10 points and smiley face) or no reward (no points and sad face). Both possible outcomes are displayed in this example. **(B)** Reward probabilities ranged between 0 and .9 and drifted throughout the session with each option being competitive at some time during the session. **(C)** Age distribution of the final sample with dashed line indexing the age groupings cutoff at 18 years. Participants younger than 18 are referred to as adolescents; participants 18 years and older are referred to as *young adults*. Data for B and C are available in S1 Data (Figure1 tab).

The finding that the behavioural mechanisms of local and reward learning continue to change during adolescence align well with the nonhuman primate literature [6,7,14,17] and our knowledge about structural brain maturation in humans: Lateral frontal, compared to medial frontal brain regions, appear to continue to mature during adolescence well into adulthood [31,33,34], and, hence, we would expect functions that depend on this part of the brain to keep changing during this time period as well. However, only manipulation approaches can provide evidence for a causal reliance of a cognitive function on a neural substrate [16,48]. Study 2 therefore examined the impact of broad lesions to lateral frontal cortex (lesions included and/or disconnected both orbitofrontal and insula cortex) on local and global reward learning. These studies used experimental tasks and analyses pipelines that were tightly matched to study 1 in cohorts of adult patients with medial or lateral frontal lobe lesions. The results indicated that indeed intact lateral frontal cortex is causally important for both local and global reward learning. Finally, in study 3, we reanalysed nonhuman primate data [6,14] that had initially suggested that these lateral frontal regions are important for local reward learning, again using matched experimental tasks and analysis pipelines. This uncovered that lateral lesions in macaques (that likely disconnected both orbitofrontal and insula cortex) also impaired global reward learning. This offers new insights into how the GRS guides choices

differently in humans and macaques. While humans showed negative GRS effects, macaques showed positive ones. Together, our results suggest that local and global reward learning mature during adolescence (study 1) and that both learning mechanisms causally depend on (subregions within) lateral frontal cortex (study 2 and study 3). This suggests that the protracted neural maturation in lateral frontal regions [31,33,34] is a key driver for the maturation of local and global reward learning during adolescence.

## Results

### Study 1: Probabilistic reward learning performance increases during adolescence

We collected developmental data from human participants on a well-characterised 3-choice probabilistic decision-making task (Fig 1) adapted from paradigms previously used in macaques and adult humans [6,14,16,22,49]. As in past studies, here, participants made choices in an environment in which the reward probabilities varied probabilistically, and reward contingencies reversed at specific times in the task.

We first assessed developmental differences in broad measures of task performance. We found that overall task performance, as measured by total rewards acquired, increased across age. Young adults earned more total rewards than adolescents (independent samples $t$ test, $t_{386}$ = 3.47, $p$ = 0.001; Fig 2A). This age-dependent difference was confirmed by a linear correlation between total rewards and age between 11 to 35 years (Pearson correlation, R = 0.16, $p$ < 0.001; Fig 2B). In accord with better overall performance, the frequency with which the highest value option (as defined by value estimates from a Rescorla–Wagner-based reinforcement learning model, see S1 Text was chosen, was significantly higher in young adults

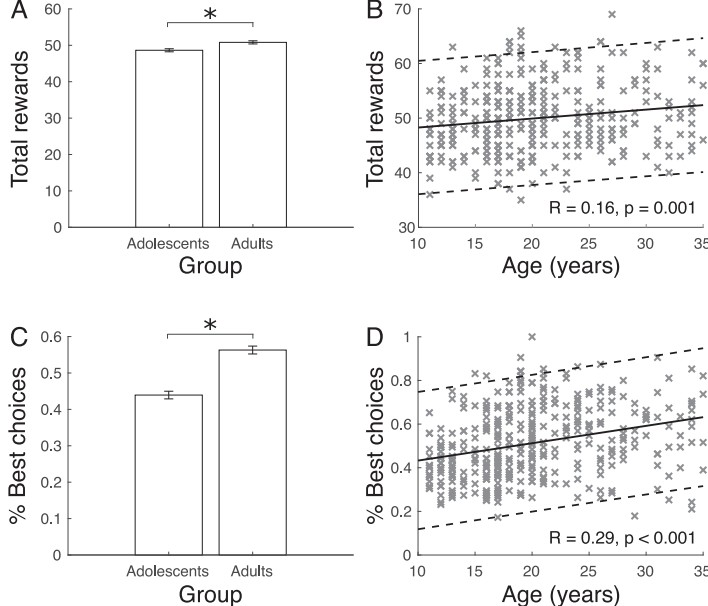

**Fig 2. Performance in probabilistic 3-choice learning task increased during adolescence. (A)** Young adults, compared to adolescents, earned more total rewards in the study. **(B)** This was also reflected in a linear increase in the total rewards earned across age. **(C)** The frequency of choosing the highest value option was higher in young adults. **(D)** It also increased across age. Note that chance performance in this 3-choice task is 0.33. ("x"s indicate individual participants; left plots show mean −/+ SEM; solid line in the right plots indicate linear trend; dashed line represented 95th% confidence interval; * $p$ < 0.05). Data for A-D are available in S1 Data (Figure2 tab).

compared to adolescents (independent samples $t$ test, $t_{386} = 7.89$, $p < 0.001$; Fig 2C) and correlated with age (Pearson correlation, R = 0.29, $p < 0.001$; Fig 2D). Follow-up model fits suggest that the relationship of age with total rewards and percentage best choices were best characterised by a quadratic function (Table A in S1 Text).

## Study 1: Local and global reward learning change during adolescence

To dissociate local and global reward learning, we used an established general linear model (GLM) approach (Methods) originally developed for the study of nonhuman primates [7]. The analysis captured the temporal dynamics of learning by analysing participants choices in a reference frame of "stay" versus "leave" behaviour. For each trial t, we observed participants' choice C and quantified their tendency to either stay with or switch away from that choice C on trial t + 1. In this "credit assignment GLM," we simultaneously accounted for several factors driving choice (Fig 3A). This allowed us to discern whether the observed changes in general task performance were driven by specific subcomponents of learning: the previous local rewards that were delivered for choosing C (CxR-history or local reward learning), the pure choice history (C-history) reflecting a tendency to repeat choices irrespective of reward receipt, and, importantly, the GRS, which reflects the overall previous reward history irrespective of choice.

First, we examined the effects of local reward learning (CxR-history) across our entire sample (regardless of age). As expected, the effects of local reward on choices differed by time point (1-way ANOVA: $F_{3,1035} = 76.42$, $p < 0.001$) with the most recent local reward at time point t ($CxR_t$) having a significantly larger effect than the previous ones, even after Bonferroni correction (for all pairwise comparisons of $CxR_t$ using paired $t$ tests: $t > 9.095$, $p < 0.001$). When a chosen option was rewarded, then there was an increased tendency to stay with the option and choose it again (one-sample $t$ test; $t_{352} = 10.92$, $p < 0.001$). Comparing the effect sizes of $CxR_t$ between adolescents and young adults showed that the size of this effect was bigger in young adults (independent samples $t$ test, $t_{351} = 4.34$, $p < 0.001$; Fig 3B) suggesting increasing associability between rewards and local choices. Correlation analyses showed a significant positive relationship between age and $CxR_t$ (Pearson correlation, R = 0.22, $p < 0.001$; Fig 3C), which follow-up model fits suggested was best characterised by a linear function rather than a quadratic one (Table A in S1 Text). By contrast, we found no developmental changes in reward-unrelated C-history effects (Fig A in S1 Text).

Next, we examined the influence of the GRS on staying with a currently pursued choice using the same GLM model reported above: This assured that any identified GRS effects were dissociated from those of local reward learning. GRS was calculated by averaging recent rewards irrespective of choice and nonzero effects indicate that the overall average levels of rewards influence decisions to stick with a choice. Previous work has shown that GRS effects are positive in macaque monkeys [7]. However, in our human sample, strikingly, we found a significantly *negative* effect of the GRS (one-sample $t$ test on all participants, $t_{352} = 7.00$, $p < 0.001$). The effects were significantly negative in both the adolescent (one-sample $t$ test, $t_{154} = -2.78$, $p = 0.006$) and the young adult sample (one-sample $t$ test, $t_{197} = -6.72$, $p < 0.001$; Fig 3D). That indicates that irrespective of directly reinforced choices, if participants had observed many rewards in the recent past (high GRS), then they were more likely to switch away from the current choice. By contrast, if the GRS was low, indicating the absence of better alternatives in the past, then participants were more likely to continue pursuing their choice even in the absence of local reward. Importantly, we predicted that if GRS effects are indeed mediated by late maturing regions of cortex, they would change during adolescence. In accordance with our prediction, we found that the GRS effect was more negative in young adults

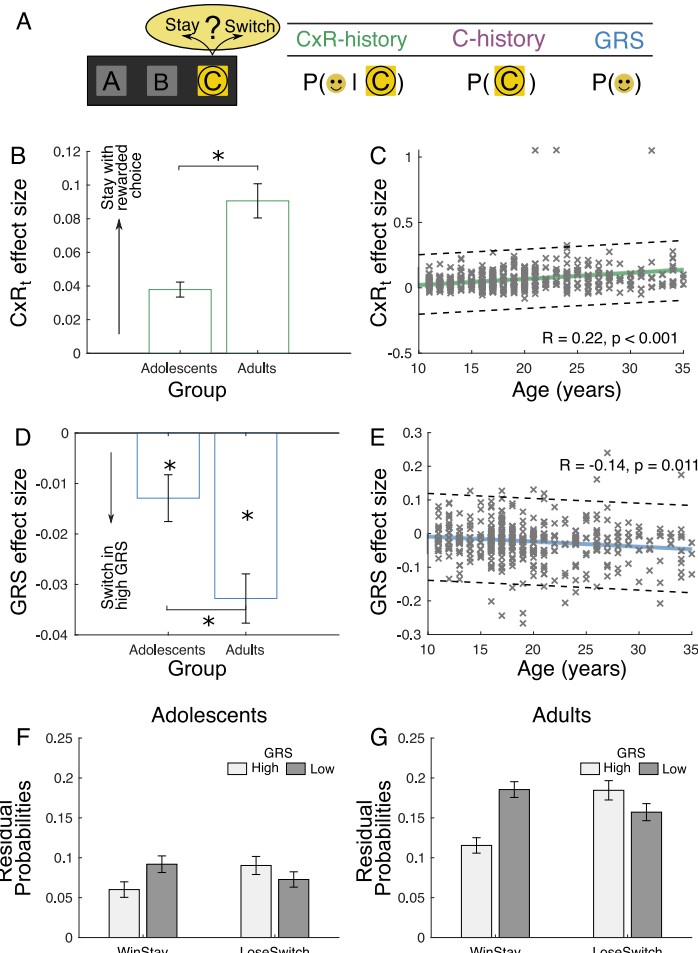

**Fig 3. Local reward learning and GRS-based learning became more pronounced over the course of adolescence.**
**(A)** In our "credit assignment GLM," we reframed the 3-choice decision problem as a foraging-style decision between staying and switching away from a currently pursued choice C. For every trial, we considered the chosen option C and analysed whether participants would stay with C on the next trial. We analysed this decision as a function of three sets of regressors: previous local (i.e., contingent) rewards for C (CxR-history), the pure choice history (C-history), and the global reward state (GRS). The right-hand illustration indicates the quantities that encapsulate these three effects: the reward outcome (schematized by a smiley face) contingent on a choice (i.e., the probability of reward given C), the repetition of a choice per se (i.e., the probability of choosing C independent of reward), and the overall recent reward probability irrespective of choice (i.e., the probability of reward independent of C). Panels B, C, D, and E show effect sizes for component parts of this GLM. **(B)** Considering the effect of the most recent outcome on the tendency to repeat a choice ($CxR_t$), we found that young adults had a significantly stronger tendency to repeat rewarded choices compared to adolescents. **(C)** The effect size linearly increased with age. **(D)** Independent and in addition to local reward learning, the GRS had a negative effect on staying with an option: Participants tended to stick more with a choice if it was encountered in the context of a low overall GRS. Such negative GRS effects were found in both adolescents and young adults with a significant difference between them. This indicates that young adults, even more than adolescents, had the tendency to contextualise rewards by the GRS. **(E)** This was replicated by a linear decrease of GRS over time. **(F)** Plot shows residual probability of staying after a win and switching after a loss (i.e., a no-reward) separated by low and high GRS (median split) for adolescents. **(G)** The same is shown for young adults. Note that in this visualisation, the GRS main effect from panels B and E is expressed as an interaction with WinStay/LoseSwitch strategy in panels F and G. The interaction effect increased for older participants: Participants were even more likely to repeat rewarded choices when encountered in a low GRS (darker bars) and, simultaneously, more likely to switch away from losing choices if encountered in a high GRS (lighter bars). ("x"s indicate individual participants; plots show mean −/+ SEM; solid lines in the right plots indicate the linear trend. Dashed lines represent 95[th]% confidence intervals. $^*p < 0.05$). Data for B-G are available in S1 Data (Figure3 tab).

compared to adolescents (independent samples $t$ test, $t_{351} = -2.89$; $p = 0.004$; Fig 3D) and correlated negatively with age (Pearson correlation, R = −0.14, $p = 0.011$; Fig 3E). Again, followup model fit analyses suggested that this relationship was best characterised by a linear function rather than a quadratic one (Table A in S1 Text).

Note that in contrast to the developmental changes in local and global reward learning–computations linked to lateral orbitofrontal and anterior insula cortex, we found no evidence for developmental changes associated with some decision variables previously associated with medial orbitofrontal/ventromedial prefrontal cortex. We considered two markers of decision computations: (1) the decision noise as calculated with a reinforcement learning model; and (2) a "bias by irrelevant alternatives" effect. Both have been related primarily to medial orbitofrontal/ventromedial prefrontal cortex functions in the past [5,7,14,16] and found neither showed developmental changes across the age range tested, potentially suggesting that these have already reached a relatively stable functional maturation point by adolescence (Fig B in S1 Text).

Our results suggest that the GRS alters the behavioural response to rewards received for a current choice over and above the effect of local rewards. To illustrate the effects of the GRS more directly, we plot choice residuals as a function of the GRS and the most recent local reward, $CxR_t$, and age using a $2 \times 2 \times 2$ ANOVA. We rearranged the data as a function of winStay (staying with a choice after a local reward at time point t) and loseSwitch (switching away from a choice after a negative local outcome at time point t; see Methods). The analysis revealed an interaction of winStay/loseSwitch and GRS independent of age group (winStay/loseSwitch × GRS interaction, $F_{1,380} = 71.69$, $p < 0.001$) illustrating the GRS effect observed before: While participants were more likely to stay after a reward, they did this even more in a low GRS; in a high GRS, they were quicker to switch away from unrewarded choices. However, critically, the GRSxWinStay/loseSwitch interaction changed with age group in a manner suggesting that adolescents were relatively less influenced by the GRS in value updating (winStay/loseSwitch × GRS x age: $F_{1,380} = 7.97$, $p = 0.005$). Older participants, by contrast, showed a stronger contrast effect after receiving reward: In low-GRS environments, they were particularly likely to stay with rewarded options and less likely to switch away from unrewarded ones.

Notably, such a negative directionality of the GRS effect is in line with theoretical predictions from behavioural ecology [50] and suggests that to maximise rewards over the long run, reward outcomes should be referenced to the background rate of reward available in an environment: Animals should spend longer foraging for reward if alternative options are scarce, whereas they should be quick to abandon a depleting food source if the frequency of highvalue alternatives are high. By conceptualising participants' choices as stay/leave decisions, we were able to identify precisely this choice pattern in our human participants in a 3-option bandit task: A negative GRS effect meant that participants switched away from an option more readily when high-value alternative options were available and they persisted with poor options when the value of the alternatives were low [50–55].

Interestingly, negative GRS effects and positive CxR effects were negatively correlated across participants (Pearson correlation; R = −0.16, $p = 0.002$; Fig C in S1 Text) and both mechanisms correlated with broad task success. Independent of age, there were significant positive correlations between local reward learning and the total rewards earned on task (r = 0.244, $p < 0.001$) and proportion of best choices (r = 0.543, $p < 0.001$). This pattern was mirrored for global reward learning with a negative correlation with total rewards earned that trended towards significance (r = −0.10, $p = 0.069$) and a significant negative relationship with proportion of best choices (r = −0.17, $p = 0.001$). This pattern of results indicates that participants who performed particularly well in linking local rewards with the specific choices that caused them also had more negative GRS effects. This suggests that both aspects of reward

learning, local assignments of reward and the ability to switch away from unrewarded choices more easily if the reward environment was rich, constituted complementary aspects of task-adaptive behaviour with both processes significantly and simultaneously gaining more influence over behaviour during adolescence. Importantly, our GRS effects of interest also remain stable when varying the history length over which the GLM is calculated (Fig D in S1 Text).

Note that the effect of local reward learning/contingent reward learning in our GLM is conceptually similar to a learning rate fitted with a reinforcement learning algorithm. Both denote the weight that a new outcome has for updating the value of the corresponding choice [7,56,57]. Higher learning rates, just as a higher local reward learning effect sizes, indicate that an outcome changes the future value of a choice more strongly. Correspondingly, there is a strong positive relationship between the learning rate fitted from our reinforcement learning model and local reward learning (r = 0.35, $p < 0.001$; correlation of learning rate with $CxR_t$). By contrast, the GRS effect is conceptually different from a reward learning rate, because it indicates the effect of a longer-term average reward that is not specifically linked to a choice. Hence, reinforcement learning rate and GRS effect size are uncorrelated (r = −0.05, $p = 0.340$). The GRS effect therefore indicates a qualitatively different effect. Also as expected, neither local nor global reward learning were associated with the inverse temperature from the reinforcement learning model, as the latter indices decision noise rather than the weighting of reward outcomes (inverse temperature versus $CxR_t$: r = 0.07, $p = 0.168$; inverse temperature versus GRS: r = −0.04, $p = 0.494$).

## Study 2: Local and global reward learning are impaired by lesions to lateral frontal lobe

The finding that the behavioural mechanisms of local and reward learning continue to change during adolescence align well with the nonhuman primate literature [6,7,14,17] and our knowledge about structural brain maturation in humans. Specifically, compared to medial frontal brain regions, lateral areas appear to continue to mature during adolescence well into adulthood [31,33,34]. Hence, we would expect functions that depend on this part of the brain to keep changing during this time period as well. Our own analysis of Human Connectome Project data [58,59] in a set of selected reward-sensitive regions of interests (ROIs) confirmed that the greatest age-related differences over our investigated age range existed in lateral and not medial regions of the brain's reward circuitry (Fig E in S1 Text). However, only manipulation approaches can provide evidence for a causal reliance of a cognitive function on a neural substrate. Study 2 therefore examined the impact of broad lesions to lateral frontal cortex on local and global reward learning. In study 2, we reanalysed behavioural data in adult patients with lateral ($n = 4$) and medial ($n = 4$) frontal lesions using the same experimental paradigm as study 1 (Fig F in S1 Text [16]) and a matched analysis pipeline. As is often the case in patient lesion studies, the lesions did not adhere to strict anatomical boundaries. The lateral lesions encompassed regions related to local reward learning in lateral orbitofrontal cortex [6,14,17] as well as more posterior regions in the anterior insula linked to global reward learning [7]. While this was a convenience sample, as the data already existed, the developmental behavioural task was designed to specifically align with these previously published experimental paradigms. We also used the same "credit assignment GLM" employed in study 1 (Methods).

We compared lateral frontal lobe patients to a brain damaged control group of patients with lesions to the medial frontal lobe. We would expect participants with lateral lesions to rely less on local rewards (decreased $CxR_t$ effect) and also to exhibit a less negative GRS effect compared to subjects with medial lesions. In other words, we would expect a "lesion site" [lateral, medial] by reward type [$CxR_t$,GRS] interaction. This was indeed precisely the effect we found

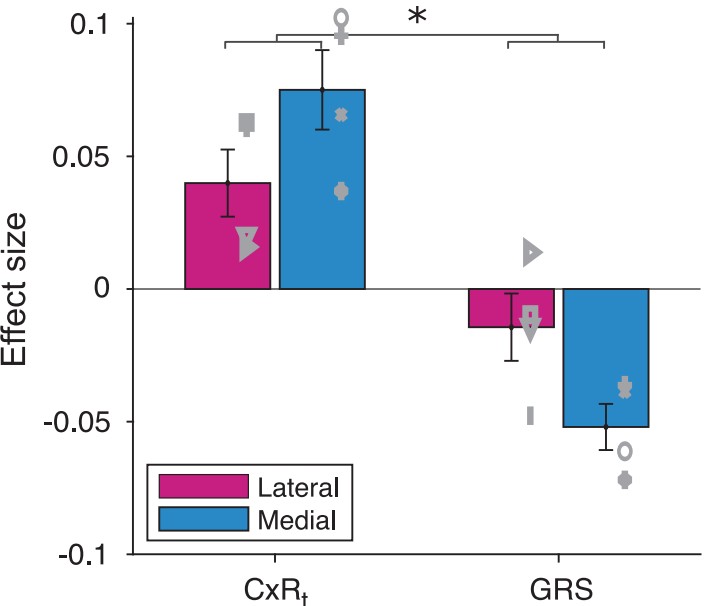

**Fig 4. Lateral frontal lobe lesions impair both local and global reward learning.** Compared to Medial patients (blue), lesions to the lateral frontal lobe (pink) in adult humans reduced both local and global reward learning. This was apparent by a simultaneous reduction of the $CxR_t$ effect sizes and a less negative GRS effect ($^*p < 0.05$; symbols indicate individual patients). Data for this figure are available in S1 Data (Figure 4 tab).

($F_{1,6} = 7.4$; $p = 0.035$; Fig 4). Lateral frontal lobe lesions caused patients to rely less on local rewards when learning about their choice options and at the same time their learning was less influenced by global reward learning. This suggests that lateral frontal cortex is the likely neural substrate that enables intact local and global reward learning.

## Study 3: Macaque lateral frontal lobe lesions change the impact of contingent rewards and the GRS on choice

Finally, we reanalysed the nonhuman primate data that had initially contributed to the suggestion that a subregion of the lateral frontal lobe, i.e., the lateral orbitofrontal cortex, is causally important for credit assignment and local reward learning [6,14]. We did this for two reasons: first, to follow up and confirm an intriguing effect in our data using more closely matched tasks: GRS effects were negative in the human sample reported here, while they were positive in previous macaque work [7]. This meant macaques stayed with a choice more when the GRS was high [7], whereas human participants reported here switched more when the GRS was high. The second reason was to examine, for the first time, whether changes in global reward learning were also apparent after lateral frontal lobe lesions in macaques, like in our human lesion data. We combined our "credit assignment GLM" with several previously published data sets of macaque choice behaviour [6,14]. This allowed our analysis to be optimised towards discovering fine-grained effects of the GRS in a uniquely large data set. We used linear mixed effects (LME) models to account for the fact that multiple sessions belonged to the same individual. We analysed 190 sessions from intact monkeys, 45 sessions from monkeys with lateral prefrontal lesion, 55 sessions from monkeys with medial prefrontal lesion, from an overall of 7 monkeys aged 4 to 10 years. Note, while these lateral lesions targeted orbitofrontal area 11 +13, there is strong reason to believe that they also disconnected lateral area 47/12o and likely other neighbouring regions including anterior insula cortex (see [18,60] for discussion and

refer to Methods for specifics of the lesions sites and nomenclature). In other words, just as in our human lesion study, we must assume that multiple subregions of the lateral frontal lobe that have dissociable functions were affected by the lateral lesions.

Controlling for local reward learning, we found a small but significantly *positive* effect of the GRS on stay decisions in the baseline data (intercept-estimate = 0.007, SE = 0.003; $\chi 2(1)$ = 5.122, $p$ = 0.024; Fig 5A). Therefore, we indeed found a sign-reversed effect of the GRS in macaques compared to our human participants using matched experimental paradigms and the same "credit assignment GLM." Moreover, comparing the effects of lateral frontal lobe lesions to medial lesions in macaques, we found that lateral lesions, too, significantly impacted global reward learning. This mirrored the findings from our human lesion study. However, strikingly and in contrast to our human sample, the GRS effects we observed after lateral lesions were significantly *stronger* (rather than weaker) compared to medial lesion groups (estimate-lateral = 0.027, SE = 0.009; $\chi^2(1)$ = 5.080, $p$ = 0.024; Fig 5B). This finding further strengthens the idea that both species use the GRS qualitatively differently during learning within the context in which these experiments were conducted. While humans rely negatively on the GRS and this capacity is abolished after lateral lesions, positive GRS effects are amplified in macaques after lateral frontal lobe lesions. This supports the contention that the GRS effect reflects a task-adaptive process in humans, which matures during adolescences and is compromised by lesions, whereas in monkeys, the GRS effect may lead to a suboptimal "spread" of reward, which is even increased by lateral frontal lobe lesions (Fig 6).

Collectively, the findings from study 2 and study 3 suggest that lesions to the lateral frontal lobe, which included lateral orbitofrontal cortex and likely disconnected the neighbouring anterior insula cortex, causally impacts both local and global reward learning. This result corroborates the idea that both reward learning processes rely on closely adjacent neural substrates in lateral frontal lobe. It further suggests that the extensive grey matter changes observed in lateral orbitofrontal and anterior insula cortex during adolescence [31,33] (Fig E in S1 Text) are likely anatomical correlates of the participants' increased capacity for both local and global reward learning across adolescent development.

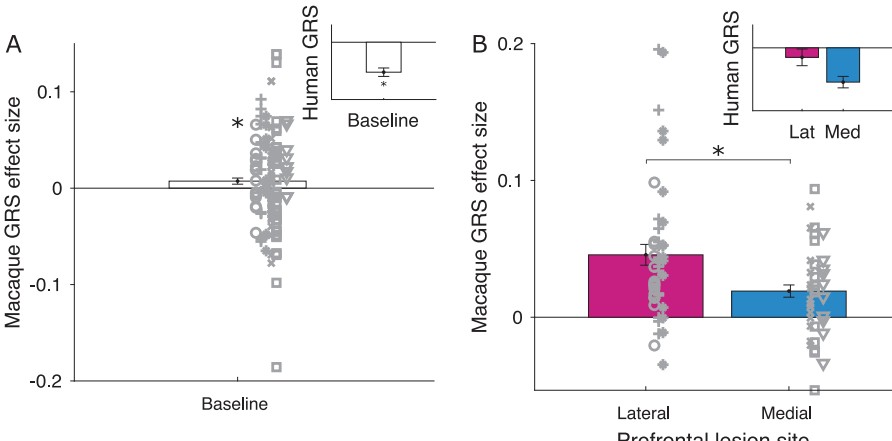

**Fig 5. Positive GRS effects in macaques increase after lateral frontal lobe lesions. (A)** In intact monkeys, we found a small but significantly *positive* effects of the GRS on stay decisions, which was sign-reversed relative to the negative GRS effect we had found in humans. The inset shows the human GRS effect averaged for all ages (11–35 years) in the matched experimental paradigm (see Fig 3, study 1). **(B)** We compared the effects of frontal lobe lesions in the macaque monkey and revealed that the positive GRS effects after lateral frontal lesions were significantly *stronger* compared to medial lesion groups. The pattern mirrored the human lesion results but in the opposite direction, with lateral lesions in humans abolishing the negative GRS effect (inset: GRS effects from Fig 4, study 3; "Lat" abbreviates "Lateral," and "Med" abbreviates "Medial") (*$p$ < 0.05). Data for A and B are available in S1 Data (Figure 5 tab).

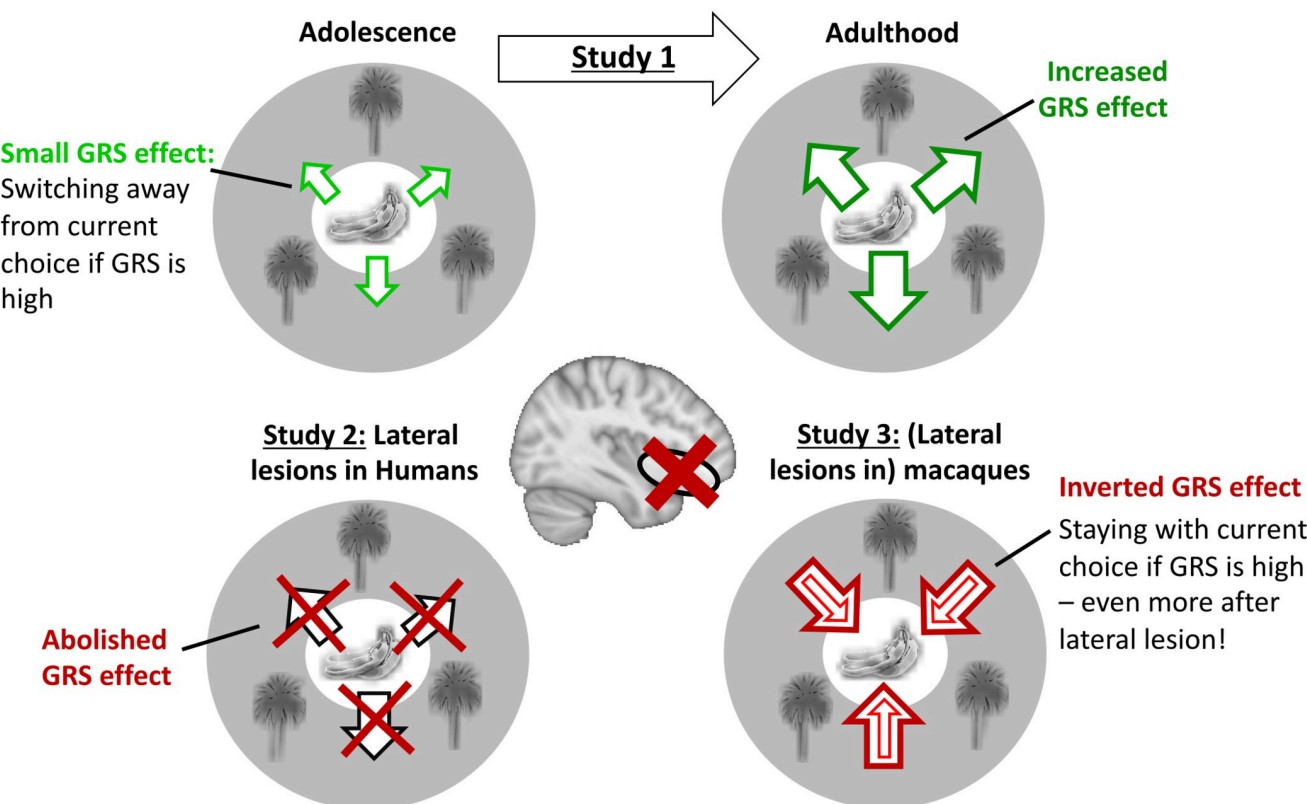

**Fig 6. Conceptual summary of GRS effects across studies. Study 1: Upper two panels**. People must behave adaptively in complex reward environments. They pursue a current choice (banana symbol within circle) that is embedded in the GRS—the global levels of reward afforded by the environment over time (tree symbols on the periphery of the circle). Adolescents switch away from the currently pursued choice if the GRS is high (small arrow pointing outwards). This can be understood as a contrast effect comparing choice and GRS. Adults show such a contrasting effect of the GRS even more strongly. They contextualise the current choice within the set of alternative options. Knowledge that rich alternatives exist makes adults switch away from their current choice more easily. The current choices appear less valuable if the GRS is very high. The increased reliance on the GRS over the course of development coincides with grey matter maturation in lateral frontal lobe regions including the anterior insula and lateral orbitofrontal cortex. **Study 2: Lower left panel**: Lesions to lateral frontal lobe (affecting multiple subregions) reduces the GRS effect in human adults. **Study 3: Lower right panel**: Macaques also contextualise current choices within the GRS. However, macaques use the GRS fundamentally differently compared to humans. They show "spread of effect": The GRS positively affects the value of a current choice, and this makes macaques stay more with a choice if the GRS is high. Strikingly, lesions to lateral frontal lobe (again affecting multiple subregions) *increase* rather than decrease this effect (thick arrow surrounding small arrow indicates stronger GRS effect after lesions to lateral frontal lobe).

## Discussion

We investigated the development of component processes of reward learning that have been linked to neighbouring regions of orbitofrontal and anterior insula cortex in studies of nonhuman primates: local reward learning (or "contingent credit assignment") and noncontingent global reward learning based on the GRS [6–8,14,17,18]. These reward-related brain regions have a particularly protracted maturation profile and continue to change well into adulthood [31,33,34] (Fig E in S1 Text). Therefore, we tested whether cognitive functions that are likely to depend on these regions keep changing during this time period as well. Indeed, we have shown that both local and global reward learning matured across development. We showed that participants' decision to switch or stay with the current choice was positively influenced by local reward that was received for making a specific choice and negatively influenced by the GRS. These mechanisms increased in their respective influence across adolescent development (study 1; Fig 3). In contrast, we found that reward-guided decision mechanisms linked to more medial frontal lobe regions did not show developmental differences over the same age

range (Figs A and B in S1 Text). However, only manipulation experiments such as lesion studies can reveal a causal relationship between a neural substrate and a cognitive process. Therefore, we conducted two lesion studies—one in humans (study 2) and one in macaques (study 3)—that assessed the impact of lesions to broad parts of lateral frontal cortex (likely affecting both the anterior insula and lateral orbitofrontal cortex; see below) to local and global reward learning. The experimental paradigm was a 3-armed bandit task (Fig 1) and was closely matched across studies. We used the same "credit assignment GLM" [7] in all studies to ensure that all three studies measured local and global reward learning in the same way. Both lesion studies showed that lateral prefrontal cortex is indeed causally necessary for intact local and global reward learning in both species (Figs 4 and 5). This suggests that structural changes in lateral parts of prefrontal cortex underlie the developmental changes we observed in behaviour. Strikingly, humans and macaques differed in the way they were guided by the GRS. Humans used the GRS to "contrast" it with the current choice and were likely to switch away from a choice if the GRS was high [51,55,61,62]; macaques showed "spread of effect" [6,11,15] and were more likely to stay with their choices if the GRS was high. Lesions to lateral frontal cortex altered the GRS effect in both species. However, while it abolished the negative GRS effect in humans, it increased the positive GRS effect in macaques (Fig 6).

These results suggest that over development humans are increasingly influenced by local and global reward states in their decision to switch or stay with their current choice (Fig 3). The increased influence over development of the local reward learning mechanism is particularly interesting in the context of its proposed evolutionary adaptive role in reducing costly errors in uncertain and changeable environments, compared to competing striatal-based reinforcement-learning systems [63]. Compatible with previous work, which has shown reduced contingency learning abilities in young children [64], and impaired updating of stimulus–reward associations from probabilistic feedback [65], here, we demonstrate that these mechanisms continue to develop into early adulthood. Critically, we also observed differences in how humans at different ages use the GRS to contrast new rewards with the baseline level of rewards encountered in the past. More broadly though, such a process can support adaptive choice switching and exploration [52,53]. Consistent with this idea and highlighting the utility of a negative influence of the GRS, we found that participants that are strongly influenced by the GRS are also more influenced by local reward learning (Fig C in S1 Text) and perform better, further suggesting complementary neural substrates that contribute to cognitive and behavioural flexibility. These findings may provide additional avenues towards understanding developmental changes in attitudes towards exploration, risk, and uncertainty from a mechanistic perspective [66–68] that have previously been interpreted as differences in feedback monitoring, inhibitory and cognitive control, and risk-taking. Indeed, this may help explain mixed developmental findings in which some studies report increases in risk tolerance between adolescents and adults, while others find no differences [69–73]. For instance, our results indicate that adolescents may display stronger persistence with unrewarded options in cases when the GRS is high. By contrast, young adults may more readily switch away from an unrewarded choice as the high GRS discourages exploring new choice options and incentivizes switching back to previously rewarded options. This weaker reliance on the GRS in adolescents may translate into increased persistence with bad choice options, risk-taking, and may help explain adolescents' greater tolerance of uncertainty [71,74–76]. However, note that learning processes in adolescents, compared to older people, differ in style and not only in terms of optimality [42,77,78]. For example, there is a shift from model-free mechanisms to model-based and counterfactual learning strategies [45,79] across adolescence. Importantly, global reward learning differs from model-based learning mechanisms [80,81] in that no knowledge about state relationships is needed and its anatomical substrates appear distinctly tied to

anterior insula [7,82]. However, in a similar manner to the shift towards model-based strategies [80], the benefits of negative GRS effects, just as the ones of increased local reward learning in our older participants, might turn out to be adaptive only in environments where exploration is relatively discouraged. In such instances, choices should be directed towards options with high values at the expense of sampling more uncertain options that nonetheless might prove more beneficial in the long run [23,83,84]. Indeed, the GRS may be dynamic and dependent on the structure of the reward environment. In the current experiment, reward schedules across all three studies were probabilistic and variable. In more blocked designs, the GRS may be less informative than in quickly changing environments, and so be less influential on the current choice.

Our results also contribute to the debate about the development of reinforcement learning [9,44,85]. Studies indicate that overall, the learning rate, i.e., the speed of updating the value of a choice, increases during adolescence [41–43,43–46]. We find the same in our study 1 (Fig B in S1 Text). Indeed, the increase in local reward learning in our "credit assignment GLM" could be interpreted along similar lines—as an increase of the weight that an outcome has on changing an option's value. The strong positive correlation between the learning rate from our reinforcement learning model and local reward learning effect size is a further indication of this. However, studies have begun to examine increases in reward learning rate in more detail, and, as highlighted above, the particular task context plays a big role in whether increased learning rates are observed and if they are desirable to optimise rewards [42]. Another consideration is that learning from outcomes might differ depending on whether that outcome is positive or negative, although these effects, again, appear context-dependent [43,45,86,87]. Our findings that the GRS exerts an increasingly negative effect during development adds to these ideas and highlights influences on reward learning that go beyond changes in a unitary reward learning rate. GRS effects were unrelated to a simple reward learning rate in previous work [7] and also in our current data set. Instead, they contextualise a current choice based on the global reward environment. This mechanism can add to the changes in value observed for a choice and can, in effect, lead to different effective learning rates for positive and negative outcomes [7,52,53]. The negative GRS effects observed here predict higher learning rates for positive outcomes if the GRS is low, and higher learning rates for negative outcomes if the GRS is high. A promising avenue for future research would be to follow up on these predictions and conduct a more formal modelling analysis of the developmental GRS effects. It might help explain diverging results by suggesting that analyses of reward learning rates should take into account the global reward levels present in the experiments. Neurally, our results suggest that subregions within lateral frontal cortex, specifically anterior insula and orbitofrontal cortex, are particularly promising target regions to look for neural correlates of the development of reinforcement learning mechanisms. These subregions have been shown to integrate rewards with different time constants in adults [7,61,82], and, in adolescents, higher learning rates for negative outcomes are linked to greater activity in the anterior insula [43].

Our hypothesis that local and global reward learning would increase during adolescence was very much guided by studies of human brain maturation, suggesting that lateral parts of prefrontal cortex mature later than medial ones [31–33]. Indeed, a selective analysis of complementary HCP imaging data indicated that lateral area 47/12o, as well as the anterior insula, showed a longer developmental maturation profile compared to the medial orbitofrontal/ventromedial prefrontal cortex, anterior cingulate cortex, and amygdala. These regions showed a significant decrease in grey matter across adolescence that continued well into young adulthood (Fig E in S1 Text). Lateral orbitofrontal cortex and the anterior insula cortex are strong potential candidates to underlie the behavioural differences in local and global reward learning seen across the same period of development. While this conjecture is indirect, it is well known

that localised regional grey matter volume correlates with motor, cognitive, and social skills [18,27,88,89]. Indeed, we recently demonstrated in macaques that grey matter around the principal sulcus is causally altered by extended training in discrimination reversal learning, with grey matter variation in this region related to individual variation in training speed [18]. Longitudinal studies examining grey matter maturation and the development of reward learning in the same sample are needed to provide more direct evidence about this link between cognitive and neural development.

However, even longitudinal studies are usually correlative and as such can only provide limited evidence about causal relationships. Study 2 and study 3 therefore used finely matched experimental paradigms and analysis techniques to directly assess the causal importance of lateral frontal regions in local and global reward learning. In humans, we show that the GRS effect is reduced (i.e., closer to zero) after lateral prefrontal lesions, compared to medial orbitofrontal lesions, in combination with decreased local reward learning (Fig 4). Note that the direction of change in both local and global reward learning after lesions is highly compatible with the correlation between those two variables in the sample from study 1 (Fig C in S1 Text). This suggests that both learning processes are at least partly supported by neural mechanisms in lateral frontal cortex. In study 3, we confirm in macaque monkeys that the GRS effect is altered after lateral lesions (Fig 5). However, the macaque lesion effects appear qualitatively different. First, as we know from the same data in past work [6,14], local reward learning is reduced after lateral lesions. This means that both humans and macaques show a decrease of contingent/local reward learning after broad lesions to lateral prefrontal cortex. However, rather than decreasing the negative GRS effect as in humans, lateral lesions *increased* the positive GRS effect in macaques (Fig 6). One interpretation could argue that the GRS effect, together with the decline of local reward learning, reflects a task-adaptive process in humans that matures during adolescence and is compromised by lesions. By contrast, in monkeys, the GRS effect could reflect a suboptimal "spread" of reward, which is even increased by lateral frontal lobe lesions. However, an alternative account could argue that the overall positive shift in GRS influence on choice after lesions in both humans and monkeys reflects a general role for the lateral frontal cortex in contextualising the GRS to avoid or suppress the influence of spread of effect mechanisms. In humans, this suppression is strong enough to produce a negative GRS effect, but it is less influential in macaque choices.

In all analysed macaque data sets here and consistent with previous work on the GRS [7] and credit assignment [6,17], in macaques, the effect of the GRS on choice was *positive*. This contrasted with the negative GRS effect in human participants (study 1 and study 2). This potential species difference is striking, particularly considering that we used a variant of a widely used probabilistic learning task that was matched across studies. However, species comparisons are inherently difficult to interpret. For example, despite the matched tasks, clear differences persisted in the way subjects were introduced to the study (verbal instructions versus weeks of training) and the setting in which the experiments were conducted. Nevertheless, one interpretation of the observed GRS differences is that human behaviour was more in line with ideas from optimal foraging theory, which suggest that a value's choice should be contrasted with the background reward rate of the environment [50,51,55,61,90]. This can promote optimal choice switching and exploration [52,53,55]. However, this view assumes that participants treated the trials in the experiment as discrete, unrelated instances. In contrast, the positive GRS effect in macaques might suggest that nonhuman primates do not perceive the task as a series of discrete and unrelated trials. Instead, they might expect intertrial contingencies. For example, macaque might assume that an action on trial n has an influence on the outcome that is received on trial n+1 (which is not the case; only the action on trial n+1 determines the outcome of trial n+1). A positive GRS effect indicates that, in line with these considerations,

reward on trial n can increase the unrelated choice that is made one trial later, on trial n+1. Such positive GRS effects are therefore not optimal for this task. However, it can be beneficial in environments that do have such dependencies across trials. Often natural environments are structured in multistep action sequences [91], and in such a setting, positive GRS effects might be adaptive.

However, it is also important to acknowledge that our lesion results are spatially limited in the precision with which they can pinpoint the functional roles of the anterior insula as the lesion in both species are either relatively unspecific (in the case of human patients) or as a likely result of disconnected fibres of passage (in the macaques). Despite this, there are several reasons to believe the GRS effects localise to the anterior insula. First, macaque anterior agranular insula BOLD signals encode the GRS strongly and bilaterally [7,82], and human anterior insula also carries similar reward signals [61]. Furthermore, it is the bilateral agranular insula that undergoes the most profound grey matter volume changes when training macaques in reversal learning tasks such as ours [18]. Therefore, anterior insula cortex and lateral orbitofrontal cortex are likely to harbour complementary reward learning computations that jointly mature during adolescence as they gain influence over learning and choice.

Lateral prefrontal regions, beyond orbitofrontal cortex, are also all late to mature across adolescence [33,92]. The more dorsolateral prefrontal regions are associated with intelligence, fluid cognition, working memory, and attentional control [93–95]. A critical direction for future work will be to examine the interactions between these developing cognitions, brain network dynamics, and the learning mechanisms described here. Recent advances in network neuroscience offer exciting methods to characterise individual differences in complex cognitions as a function of local and global brain network topology and community structure [96–98]. Characterising these interactions could ultimately improve predictions of transdiagnostic features of neurodevelopmental and behavioural trajectories.

In summary, our multimodal approach suggests that lateral frontal cortex is a particularly dynamic locus of neural maturation driving cognitive changes in both local and global reward learning during adolescence and into young adulthood. Evidence of heterogeneity across the developmental profiles of the reward-guided component processes and the underlying neural network highlight the importance of understanding and quantifying the development of the whole prefrontal cortex at a functionally meaningful resolution. Future longitudinal studies should examine multimodal changes in lateral orbitofrontal and anterior insula cortex and the respective parallel changes in the adaptive influence of local and global reward learning. Understanding how and why reward learning mechanisms develop across adolescence could not only begin to explain the frustrations of parents and carers of teenagers who perpetually remind adolescents to consider the consequences of their choices, but also impact their ability to adaptively learn from feedback in social, health, and educational contexts.

## Methods

### Study 1: Development of local and global reward learning across adolescence

**Participants.** Participants between 11 and 35 years old were recruited. In total, 422 participants completed the task. We refer to participants younger than 18 years as *adolescents* (i.e., ≤17 years), and we refer to older participants as *young adults*. Participants were excluded from the analysis for failing to supply age or gender data. Participants were also excluded if they only repeatedly chose one option or one location, indicated they had completed the game more than once already, or did not have parental permission. A further 7 participants were excluded from that sample as their median reaction time was more than three times the

standard deviation from the mean. This left 388 participants (260 female, median age = 19). Both adolescents and young adults were recruited via similar channels. Broad recruitment methods, including direct advertisements on local public advertising forums and social media, were used with to invite young adults and adolescents (via their parents). Adolescents were additionally recruited via their parents through contact with local schools within the Oxford-shire area. Young adults were additionally recruited via local university online advertisement and via email lists. Collaborating schools forwarded documentation inviting parents to consent to their children participating in the study by directing them to the study website. In accordance with the Declaration of Helsinki, adolescents or adults assented or consented, respectively, to participation in the study before the task began and had the option not to sub-mit their data to the study once the task and questionnaires had been completed. They were also free to quit the study at any time by closing the browser window. The study was approved by the Central University Research Ethics Committee (Project Number: R59372/RE001). Par-ticipants of all ages received no monetary compensation.

**Task and procedure.** Participants completed a 3-armed probabilistic bandit task (Fig 1A) that was modelled after a paradigm previously established in monkeys [6,14]. The task was coded in JavaScript, HTML, and CSS and hosted on JATOS (version 3.3.4). During the task, participants saw three different coloured options (blue, green, and red), which were presented in one of three locations that varied along the x-axis, with option locations randomised across trials. Clicking on one of the options resulted in either the display of a smiley face and a 10-point win or a sad face and no win (0 points). The goal of the task was to win as many points as possible. At the bottom of the screen, throughout the game, the number of points participants had won so far and how many trials they had left to play was displayed. Also, par-ticipants were specifically instructed that the "chance of winning points is different for each color" and that throughout the game, "the most rewarding color might change." The reward schedule (Fig 1B) was adapted from the one used by Noonan and colleagues [14]. Reward probabilities for each option were slowly and unpredictably drifting over time and ranged between 0% and 90%. The probabilities of each option being rewarded were independent of each other. The task was self-paced, with stimuli remaining on the screen until a decision was made and feedback was presented for 1,500 ms. After a 10-trial practice run, participants could either go back to the instructions, if they had remaining questions, or proceed to the main task, which consisted of 100 trials and took approximately 7 minutes to complete. Partic-ipants also submitted age and gender information.

### First-level analyses: "Credit assignment general linear model (GLM)"

For all behavioural analyses, we used MATLAB 2020Ra (The MathWorks) and SPSS (version 25). We applied a first-level GLM to choice data (Fig 5A; see below), which sought to under-stand the factors that influenced the decision to stay or switch from a current choice in relation to contingent, local reward assignments, choice repetition irrespective of reward, and, impor-tantly, the GRS. The latter variable captures the average recent reward levels irrespective of the specific choices that have led to reward. In monkeys, a high GRS can increase the degree to which animals stay with their currently pursued choice even if this specific choice was not rewarded [7]. We adapted the logistic GLM used in Wittmann and colleagues [7]. For every trial t, we identified the chosen stimulus C and examined whether it was chosen again on the next trial. We then tested whether such a stay/switch decision was predicted by three sets of regressors: [1] The local, contingent choice-reward history of C (CxR-history) [2]; the reward-unlinked choice history of C (C-history); and [3] the choice-unlinked reward history (GRS). The GRS regressor reflects our parameterization of the GRS and allowed us to test whether the

GRS, regardless of the choice history and the contingent choice-reward history, influenced stay/switch decisions. The regressors were constructed in the same way as in our past report [7] as follows:

1. <u>CxR-history:</u> The model captures local, contingent reward effects (CxR-history) through regressors that denote whether choices of C on trial t and also on the preceding three trials were rewarded or not ($CxR_t$, $CxR_{t-1}$, $CxR_{t-2}$, $CxR_{t-3}$). CxR-history regressors were set to 1/0 for rewarded/unrewarded outcomes. Hence, positive effects of these variables indicate that a choice is more likely to be repeated if that specific choice has received reward in the past. Note that t refers to instances in which choices of C are made and not necessarily to its presentation on consecutive trials as only the former informs the conjunctive choice-reward history of C.

2. <u>C-history:</u> The model includes three regressors to reflect the recent choice history of C (C-history; $C_{t-1}$ $C_{t-2}$, $C_{t-3}$). Irrespective of the receipt of reward, this regressor codes whether C was chosen or not, being set to 1/0 for each trial. In contrast to CxR-history, C-history captures the degree of choice repetition, i.e., the fact that past choices predict that these same choices are made in the future independent of the receipt of reward.

3. <u>GRS:</u> The model took the simple average reward on the three trials before t as an index of the overall current levels of reward. The three most recent trials were used for this in all cases. In addition, we also included the interaction of GRS with $CxR_t$ (multiplying both variables after they were normalised) to account for potential asymmetric effects of GRS and rewarded and unrewarded trials.

We applied the GLM model to the stay/switch decisions and analysed the resulting beta weights. To account for outliers, we log-transformed the beta weights and implemented an outlier rejection procedure. We only included sessions whose beta weights were within three standard deviations from the mean. From the above analysis, this led to the exclusion of a further 35 participants. Beta weights were then further submitted to a second-level analysis.

**First-level analyses: Reinforcement learning modelling and bias by irrelevant alternatives.** In a separate analysis stream, we fitted a reinforcement learning model comprising two model parameters: learning rate and inverse temperature. Specifically, we fit a reinforcement learning model with a Boltzmann action selection rule that uses information about past choices and rewards to estimate expected value for each option on each trial. The reward learning rate and inverse temperature were fitted individually to each session's data using standard nonlinear minimization procedures. These parameters, respectively, reflect the weight of influence of the prediction error and the influence of value difference of the probability of choosing an option. To account for outliers in the inverse temperature estimates, we log-normalised this parameter after fitting the model. The learning rate and inverse temperature parameters were then submitted to second-level analyses.

The reinforcement learning model formed the basis for our investigation of choice biases induced by irrelevant alternatives. This approach investigates whether choices were unduly influenced by the value of an irrelevant alternative. Methodical details are specified in Fig B in S1 Text.

**Second-level analyses.** Second-level analyses focused on age-related differences in first-level effect sizes. Because there is some uncertainty about the precise age when developmental changes in credit assignment might occur or whether they occur in a continuous fashion or stepwise, we analysed age effects in two complementary analyses. Importantly, all our key results survive both ways of analysing age effects. First, we performed an age split and

compared an adolescent subgroup (age $< 18$; $n = 159$) with a group of young adults (age $\geq 18$; $n = 228$) via independent samples $t$ tests (Fig 1C). Should developmental differences in computational subprocesses of reward learning occur during adolescence, then we should expect significant differences between the two age groups. However, we also analysed our data in a continuous way as cognitive processes may slowly mature over time irrespective of precise age boundaries. For this, we used Pearson linear correlation analyses across age.

We first used these analysis steps to perform two unique additional analyses that more broadly describe our developmental data. As initial analyses of task performance, for each participant, we calculated total rewards earned during the task. Next, we calculated the proportion of choices of the best option. This measure was derived from the estimated expected value of each stimulus option using the reinforcement learning model as described below.

Next, we used the two-step analysis pipeline for the credit assignment GLM. We analysed [1] CxR-history, [2] the $C_{t-1}$ within the C-history component (see Fig B in S1 Text), and [3] the GRS. To complement the GRS analyses described above, and in line with the analysis approach described previously [7], we conducted a follow-up analysis in our developmental data to investigate the influence of the GRS on stay/switch decisions in more detail. We estimated the residual probabilities of a choice to switch or stay by regressing out of all effects of the previous GLM, except $CxR_t$ and GRS, and their interaction. The resulting choice residuals were then binned by [1] the receipt of a reward on trial t and [2] GRS (low or high; calculated as a median split of GRS). The estimated residual probabilities derived from the subsidiary GLM investigating the influence of the GRS on switch/stay decisions were split into adolescents and adult age categories and subjected to a 2 (receipt of reward on trial t [reward; no reward]) × 2 (GRS [low; high]) × 2 (age [adolescent; adult]) repeated measures ANOVA. According to the outlier rejection procedure described above, now only a single participant was excluded from this follow-up analysis.

We then examined a linear relationship between the global reward learning (GRS) effects and local reward learning effect ($CxR_t$). For this, we calculated a correlation between $CxR_t$ and GRS, which reflects key markers of contingent credit assignment and GRS effects, respectively. As this analysis aimed to show an age-general relationship, we performed a partial correlation between the two variables controlling for age and the GLM constant. This was to ensure that this finding would not be confounded by age differences and the baseline tendency of participants to stay or switch.

Finally, we ran a series of partial correlations between the reinforcement learning parameters and local and global reward learning controlling for age. In addition to excluding subjects' beta weights from the credit assignment GLM that were three times the standard deviation plus or minus from the mean (see above; $n = 35$), we similarly applied the same exclusion criteria to subjects' learning rate and inverse temperature parameters. This resulted in an additional 7 subjects being removed from the partial correlation analysis. We also examined the utility of these mechanisms to behavioural success as indexed by the total rewards earned by each subject. Again, we used partial correlation analyses to control for the influence of age and correlated both local (indexed by the GLM's $CxR_t$ effect) and global reward learning (indexed by the GLM's GRS effect) with total rewards earned.

As a follow-up analysis, we further characterised the trajectory of our key variables of interest. We considered both linear and quadratic developmental trajectories to avoid overfitting [99–101]. Following the identification of a correlation between ages, we then fitted linear and quadratic link functions. Among these functions, we identified the one with the best fit as indicated by the lowest AIC value. These are reported in Table A in S1 Text. The purpose of these follow-up analyses was to develop a more detailed picture of the maturation of the variable of interest over our entire age range of 11 to 35 years [46].

## Study 2: Human lesions to medial and lateral frontal cortex

**Participants.**   Data from 8 adults (7 female) with focal lesions involving the medial and lateral frontal lobes were reanalysed for the purposes of the current study. Patients were originally recruited from the Cognitive Neuroscience Research Registry at McGill University to examine the impact of medial and lateral lesions on the specific influence of the credit assignment mechanism [16]. They were free from neurological or psychiatric disease and not taking any psychoactive medication. For further neuropsychological screening and demographic information, see [16]. We analysed choice data from four patients with lesions to lateral frontal lobe (3 female, mean (and SD) age 60.25 (11.4) years) and four patients with lesions to the medial frontal lobe (2 female, mean (and SD) age 61.5 (11.0) years). Age was not significantly different between the two groups ($t_6 = -0.16$, $p = 0.880$). Groupwise lesion overlap images were generated by registering patients' lesions to the MNI brain. Due to the nature of patient lesions, we refer to these lesion groups as "Lateral" and "Medial." For further details of lesion locations and cause of lesions, see Noonan and colleagues [16]. Patients were studied at least 6 months after injury (median time since injury = 6.5 years, range = 2.4 to 11.8 years). All participants provided written informed consent in accordance with the Declaration of Helsinki and were compensated for their time with a nominal fee, plus earnings based on the rewards gained in the task. The study was approved by the MNI's research ethics board.

**Task and procedure.**   Equipment, procedure, and schedules have all been fully described in Noonan and colleagues [16]. However, for completion, we will briefly describe the task and reward schedules (Fig F in S1 Text). Following instructions and practice sessions, participants played a 3-armed bandit task contextualised in terms of a free trip to the casino. During the testing session, three novel distinguishable fractal stimuli were presented on screen (Fujitsu, Lifebook T, with Windows Vista) via Presentation Neurobehavioural Systems (version 14.9). Stimulus location was computer-randomised within a triangle configuration. Participants selected a stimulus by pressing a corresponding arrow on the keyboard. A question mark at the center of the screen would disappear once the subject made a choice (Fig F(A) in S1 Text). Stimuli would remain on screen until feedback. Feedback was presented stochastically for a choice according to the reward probabilities defined by one of two schedules. Correct and incorrect feedback, a green checkmark or red cross, respectively, was presented at the center of the screen for 1,500 ms. Correct responses caused a green money bar to increase by a fixed number of pixels, tallying each subject's winnings. The participants' goal was to collect as many points as possible. Feedback was followed by a 1,000-ms intertrial interval. Participants completed two counterbalanced sessions, each with 500 trials, with new stimuli in each session and a break in between. Testing took approximately 1.5 hour to complete. Reward probabilities varied unpredictably over time and ranged between 0.1 and 1. The probabilities of each option being rewarded were independent of each other. Regardless of what the subject chose, the best option could change after approximately 25 trials (see Fig F(B) in S1 Text). Patients were tested either in a quiet room of their home or in a quiet experimental testing room at the MNI.

**First-level analyses: "Credit assignment general linear model (GLM)".**   We applied the exact same first-level GLM as detailed in study 1 but adjusted the length of the analysed trial history to account for longer-term reward history effects that might bias our results (each session had 500 trials). History trial length was extended to include 4 trials in the past in the GLM (rather than 3 trials as for study 1). Beta weights were absolute log-transformed, but no outlier procedure was introduced. Only the beta weights, which reflected local and global reward learning, were passed to the second-level analysis.

**Second-level analyses.**   Lateral frontal lobe lesion patients were compared to a control group of patients with medial frontal lobe lesions. We compared the relative influence of local

and global reward learning mechanisms, across the two lesion groups using a mixed ANOVA 2 (Lesion: lateral, medial) × 2(learning effect; $CxR_t$, GRS) design.

## Study 3: Macaque lesions to medial and lateral frontal lobe

**Subjects.**   Data from six male rhesus macaque monkeys (*Macaca mulatta*), aged between 4 and 10 years and weighing between 7 and 13.5 kg, were reanalysed for the purposes of the current experiment. These data were originally collected and analysed in Noonan and colleagues (2010) and Walton and colleagues (2010) [6,14] to examine the impact of medial and lateral frontal lobe lesions on the influence of the credit assignment mechanism. Six monkeys originally participated in an experiment reported by [6] in which three animals acted as unoperated controls, whereas the other three received bilateral aspiration lateral orbitofrontal cortex lesions following training and presurgical testing. The three unoperated control monkeys and one additional monkey who had not participated in the Walton and colleagues study then participated in [14] and were tested before and after bilateral aspiration lesions of medial orbitofrontal cortex. All animals were maintained on a 12-hour light/dark cycle and had 24-hour ad libitum access to water, apart from when testing. All experiments were conducted in accordance with the United Kingdom Animals Scientific Procedures Act (1986).

**Task and procedure.**   Apparatus, training histories, and schedules have all been fully described in [6,14]. However, for the purposes of the present study, we will briefly describe the task and reward schedules (Fig G in S1 Text). On every testing session, animals were presented with three novel stimuli that appeared in one of four spatial configurations. Configuration and stimulus position were determined randomly on each trial. Stimuli remained on screen until an option was chosen. Reward was delivered stochastically for a choice towards each option according to the reward probabilities defined by the session schedules. Stimulus presentation, experimental contingencies, and reward delivery were controlled by custom-written software. Here, we analysed data from three reward schedules employed by these studies and which formed the basis of the experimental schedules used for our human study. In these schedules, all three options were at some point competitively rewarded, and reward probabilities varied over the course of the testing session. The probabilities of each option being rewarded were independent of each other. Across different days, animals completed five sessions of 300 trials under each schedule, with novel stimuli each time. For the first two schedules, the sessions were interleaved across testing days, whereas for the last schedule, the data were run with consecutive sessions. Data were collected both pre- and postoperatively. Approximately 18 months separated testing in the Walton and colleagues experiment and training in the Noonan and colleagues study. Before testing in the latter study, all animals with medial lesions were brought to a criterion of 80% correct on three choice-reversal schedules and both preoperative groups were at roughly the same preoperative performance level as they were when they acted as unoperated controls in the former study.

**Surgeries.**   Surgical procedures and histology for the lateral and medial lesioned animals have been previously described in full in [6,14]. In brief, animals were given aspiration lesions to the lateral or medial orbitofrontal cortex using a combination of electrocautery and suction under isoflurane general anaesthesia. The lateral lesion was made by removing the cortex between the medial and lateral orbitofrontal sulci and as such predominantly targeted Walker's areas 11 and 13 but may also have included parts of area 12. Medial lesions removed cortex between the medial orbitofrontal sulcus and the rostral sulcus, mainly including Walker's area 14 but may have also included some parts of area 10. Note, the lateral aspiration lesion effects on contingency learning reported by a number of studies [6,14] have recently been argued to be caused not by cortical damage to Walker's area 11 or 13 but by the damaged cortex laying

adjacently lateral to this area beyond the lateral orbitofrontal sulcus, which transitions into ventrolateral prefrontal cortex and aligns mostly with the gyral region of the orbital part of inferior frontal gyrus [60]. This corresponds to the orbital part of area 12 (12o) in macaques and Brodmann's area 47o in humans (referred to from here as area 47/12o). The contingency learning effects are now attributed to the disconnection between areas 11 and 13 and adjacent cortex in area 47/12o [4,18]. We therefore refer to these lesions as "Lateral" and to the medial orbitofrontal lesions as "Medial."

**First-level analyses: "Credit assignment general linear model (GLM)".** We applied the exact same "credit assignment GLM" as detailed in study 1. Again, the set of regressors included local reward learning effects (CxR-history: $CxR_t$, $CxR_{t-1}$, $CxR_{t-2}$, $CxR_{t-3}$), choice history effects (C-history; $C_{t-1}$ $C_{t-2}$, $C_{t-3}$), and the GRS effect as well as the interaction of GRS with $CxR_t$. Beta weights were log-transformed, and the same outlier rejection procedure was implemented as described in study 1. From the above analysis, this led to the exclusion of 6 sessions across 3 monkeys. Only the beta weights, which reflected local and global reward learning, were passed to the second-level analysis.

**Second-level analyses.** Second-level analyses (i.e., averaging over subjects and sessions) were performed in a conceptually similar manner for monkeys and humans but differed in their implementation because of the nature of the acquired data and the goals of the analyses. We acquired several sessions' worth of data for the same macaques, whereas there was only one session per human participant.

For the macaque credit assignment GLM, we submitted resulting (outlier-corrected) beta weights/parameter estimates to separate LME models (using Matlab's fitlme) because the LMEs allowed us to account for monkey identity ("Mk") in our analyses. We grouped the data for each second-level analysis in three conditions: a baseline condition (all data that were collected in nonlesioned animals), a Lateral lesioned condition, and a Medial lesioned condition. All analyses were collapsed over experimental paradigms. We coded monkey identity as a random effect with a random intercept and random slopes for all fixed effects used in the LMEs. For significance testing of fixed effects, we used a likelihood ratio test comparing a full model with a model leaving out the particular fixed effect of interest. In addition, we report the fixed effects slope estimates and their standard errors.

We examined effects of the GRS on choice. To first demonstrate that such effects exist in our data at all, we tested whether the intercept of the LME differed from zero in separate LMEs for each lesion condition. The LMEs comprised only an intercept and the random effect of monkey identity and we compared them with LMEs without intercepts to demonstrate positive GRS effects in both conditions. The LME with intercept was constructed as follows, and this procedure was separately applied to the baseline condition and the lesion condition:

$$\text{LME1}: \text{GRS} \sim 1 + (1|\text{Mk})$$

Finally, we tested whether the mechanism by which the GRS impacts reward learning is impacted by Lateral lesions compared to Medial lesions. We therefore ran LME2 to determine differences in effect sizes between the two lesion conditions themselves (LesionType: Medial versus Lateral).

$$\text{LME2}: \text{GRS} \sim \text{LesionType} + (1 + \text{LesionType}|\text{Mk})$$

## Citation diversity statement

Recent work has identified a bias in citation practices, which results in papers from women and other minorities being undercited. The citation diversity metrics below, proposed by Zurn and colleagues [102], encourage us to proactively reflect on our citation practices, and we have taken steps to ensure a more accurate reflection of the diversity in science. By this measure (and excluding self-citations to the first and last authors of our current paper), our references contain 19.8% woman (first)/woman (last), 14.6% man/woman, 17.2% woman/man, and 48.3% man/man.

## Supporting information

**S1 Text. Supplementary Material: Fig A. Influence of choice history on switch/stay decisions does not change across adolescence**. As a control analysis, we considered developmental changes in a reward-unrelated learning mechanism that was also included in our credit assignment GLM. We examined C-history, the tendency to repeat choices irrespective of reward [1]. In general, participants were more likely to repeat the most recent choice, irrespective of reward (one-sample $t$ test; $C_{t-1}$: $t_{352} = 3.09$, $p = 0.002$). **(A)** However, $C_{t-1}$ did not differ between adolescents and adults (independent samples $t$ test; $t_{351} = 1.16$, $p = 0.245$). **(B)** Analogously, there was no correlation between age and $C_{t-1}$ ($R = 0.07$, $p = 0.191$). ("x"s indicate individual participants; plots show mean $-/+$ SEM; solid line in the right plots indicates best fitting linear trend. Dashed lines represent $95^{th}$% confidence interval). Data for B and C are available in S1 Data (Fig A tab). **Fig B. No developmental changes in decision computations**. Complementing our analyses of global and local reward learning, we also considered developmental changes in decision-related computations. **(A)** We first fitted a simple reinforcement learning model to our data. This model was fitted individually to each session's data using standard nonlinear minimization procedures and a Boltzmann action selection rule. In line with the $CxR_t$ effects reported above, learning rates for young adults were significantly higher than for adolescents (independent samples $t$ test, $t_{386} = -3.83$, $p < 0.001$). **(B)** This result was confirmed as a significant correlation with age (Pearson correlation, $R = 0.19$, $p < 0.001$). **(C)** Notably, the age groups did not differ in their general levels of decision-making noise, as the RL models' (log-normalised, to account for outliers) inverse temperature parameter did not differ with age (independent samples $t$ test, $t_{379} = -1.83$, $p = 0.068$; Pearson correlation, $R = 0.02$, $p = 0.720$). Note the effect remained nonsignificant when the inverse temperature was not log normalised. This suggests that changes in learning rates cannot be reduced to changes in decision noise. It furthermore also strongly indicates that our previous results about GRS-related maturation are not driven by differences in decision noise between the age groups. **(D)** Examining the developmental trajectory of this parameter over time also failed to reveal a significant change. **(E, F)** Finally, following our analysis approach established in human medial frontal lesion patients [2], we used a combination of multinomial logistic regression analysis and reinforcement learning modelling (see above) to examine the influence of a value-based decision bias. We considered each 3-choice decision as two binary comparisons and rearranged them such that we can extract the biasing effect of the value of a distractor option on choice. Using these expected values generated for each option on each trial, we examined whether the interactive impact of the decision-irrelevant option's value ($V_D$) on the choice between the two relevant Options ($V_X$ and $V_Y$). We applied a two-step multinomial logistic regression analysis, which has been described in full, alongside the complete set of equations, in Noonan and colleagues (2017). We chose this specific GLM to make the findings directly comparable to our previous human lesion study. In short, this approach reframes the 3-choice decision as two binary value comparisons between pairs of options.

The GLM aims to predict the proportion of choices among the three options from their expected values, with one option assigned in each decision frame as the reference category. For example, Options X and Y are the options being compared; with Option Y as the reference, Option X as the comparator, and Option D denoting the irrelevant option. Each option's values ($V_X$, $V_Y$, $V_D$) were initially derived from a reinforcement learning model described above. The present study examines distractor effects on choice as a function of potential regional differences in the speed of brain maturation during adolescence. Previous lesion studies have characterised this as a negative influence [2,3], and so we selected a model that allowed us to focus on that specific factor. The key step in the model, for the purposes of the present study, is the isolation of the contextual decision-making factor ($V_X - V_Y$)$V_D$ from the final step of the GLM outlined in Eq 1 (equation 7 in [2]). Intuitively, this term reflects the modulation of the decision variable (the value difference between the options) by the distractor.

$$\ln\left(\frac{P(X)}{P(Y)}\right) = \beta_0 + \frac{(\beta_1 - \beta_2)}{2}(V_X - V_Y) + \frac{(\beta_1 + \beta_2)}{2}(V_X + V_Y) + \beta_3 V_D + \beta_4 V_X V_Y$$
$$+ \frac{(\beta_5 - \beta_6)}{2}(V_X - V_Y)V_D + \frac{(\beta_5 + \beta_6)}{2}(V_X + V_Y)V_D \quad (1)$$

This factor allows us to examine how the expected value of the irrelevant option $V_D$ affects the comparison between X and Y (i.e., $\frac{(\beta_5 - \beta_6)}{2}(V_X - V_Y)V_D$), after controlling for the effects of the difference between the two options ($V_X - V_Y$), their total value ($V_X + V_Y$) and their interaction ($V_X \times V_Y$), as well as the independent value of the distractor ($V_D$) and the interaction between the distractors value and the relevant options' combined value (($V_X + V_Y$)$V_D$). In other words, the ($V_X - V_Y$)$V_D$ beta weight reflects the degree to which the effect of value difference between X and Y on choices between these two options was modulated by the irrelevant distractor value ($V_D$). For brevity, we refer to our variable of interest, the ($V_X - V_Y$)$V_D$, as *bias by irrelevant alternative* (BIA). In addition to the standard exclusion criteria, the regression model described below failed to fit a total of 32 participants and were excluded from this analysis. Subsequently, the factor isolated from the GLM was subjected to an outlier rejection procedure (15 participants), and the beta weights were absolute log transformed. Beta weights were then submitted to a second-level age-comparison analyses. The current choice data showed that BIA did not differ with age (independent samples $t$ test, $t_{339}$ = 1.06, $p$ = 0.291; Pearson correlation, R = 0.03, $p$ = 0.522). Therefore, in contrast to the local and global reward learning mechanisms linked to lateral prefrontal cortex, the influence of the value of third option on the binary choice may already reflect a matured functional state by the age of our sample. ("x"s indicate individual participants; plots show mean −/+SEM; solid line in the right plots indicates best fitting linear trend. Dashed lines represent 95th% confidence interval. $^*p < 0.05$). Data for A-F are available in S1 Data (Fig B tab). **Fig C. Local and global reward learning correlated across participants**. We investigated the relationships between local reward assignments and negative GRS effects. Despite the theoretical accounts arguing that a negative GRS effect might aid value learning, it could be argued that GRS effects per se are suboptimal in the context of probabilistic learning tasks. To address this, we examined the relationship of GRS with a marker of local contingent value assignment, the $CxR_t$ effect, as the latter reflects a signature of successful learning in this task. Controlling for participant age and their GLM constant, we examined the relationship between GRS and $RxC_t$ using a partial correlation. **(A)** Our findings revealed a strong negative correlation between contingent reward assignment and the global reward effect (Pearson correlation, R = −0.16, $p$ = 0.002). This suggests that individuals who are more influenced by local

reward assignment mechanisms are also more likely to rely on a negative reward contextualisation. This pattern of behaviour further supports the idea that negative GRS effects are adaptive and may co-mature with contingent credit assignment mechanisms during adolescence. Visual inspection might suggest that the correlation is potentially driven by three outliers with high contingent learning scores. **(B)** However, removal of these data points confirmed that this was not the case; instead, the correlation became even more significant (R = −0.23, $p < 0.001$). ("x"s indicate individual participants; solid line indicates linear fit). Data for A and B are available in S1 Data (Fig C tab). **Fig D. The effect of GRS on choice is stable across a broad window of reward history length and does not depend on arbitrary statistical choices**. In our main GLM, the GRS is calculated as the arithmetic mean of rewards occurring during the last three trials (see Methods). This history length was chosen a priori based on previous work and based on the number of trials in the experimental schedule. To show that the GRS effects are stable, we repeated our main GLM and varied the length of this reward history. We varied it between including only the last two trials **(A)**, the last four **(B)**, and the last five **(C)**. The respective panels show the GRS effect from these three GLMs. In accordance with varying reward history length, we adjusted the timescale of the other relevant learning mechanisms (CxR-history and C-history) in the GLM. This ensured that the GRS, CxR-history, and C-history were all calculated over the same set of past trials. Consequently, in the analysis, variance associated with one learning mechanism was unlikely to be misattributed to another learning mechanism as they cover the same duration of the trial history. For example, when extending the history length of the GRS to five trials, we also extended the history length of CxR-history and C-history by two trials. We then aggregated these alternative regression models and showed that our effects of interest remained significant. Aggregating the results across the 3 alternative choice history lengths, we compared the beta weights against zero for adolescents and adults separately in 2 one-sample *t* tests and found negative GRS effects both in adolescents ($t_{153} = -2.79$, $p = 0.006$) and adults ($t_{176} = -5.27$, $p < 0.001$). Importantly, as in our main analysis, adults have a more negative GRS effect than adolescents ($F_{1,331} = 8.14$, $p = 0.005$; main effect of age group in 2 [age group: adolescents, adults] × 3 [reward history length: 2, 4, or 5] repeated measures ANOVA). Main effects of history length or the interaction between history length and age group were not significant ($F_{2,666} = 0.578$, $p = 0.480$, Interaction $F_{2,666} = 0.711$, $p = 0.426$). **(D)** Finally, again, as in our main analysis, this developmental trajectory also manifests in a negative correlation between age and GRS effect (r = −0.19, $p < 0.001$). For this correlation, we averaged the GRS beta weights, within each subject, across the three GLMs with history length 2, 4, and 5. The average GRS beta weights were then plotted against age. Critically, these analyses all used a history length that is different from the one in the main GLM and demonstrate that our results did not depend on arbitrary modelling choices. ("x"s indicate individual participants; plots show mean −/+ SEM; solid line in the right plots indicates best fitting linear trend. Dashed lines represent 95% confidence interval). Data for A-D are available in S1 Data (Fig D tab). **Fig E. Delayed grey matter maturation in lateral orbitofrontal and anterior insula cortex relative to other networked learning and decision-making neural nodes**. Study 1 showed significant changes in local and global reward learning across adolescent development and into early adulthood. Here, we investigated the potential underlying neural changes by examining grey matter maturation in prefrontal cortex during the same time window as our behavioural sample, 11–35 years (see Supplementary Methods). We considered regions of interest (ROIs) that are related to reward processing. Local and global components of reward learning have both been previously linked to lateral orbitofrontal (lOFC) and anterior insula cortex (Ins), respectively [1,3–5]. Medial orbitofrontal/ventromedial prefrontal cortex (mOFC/vmPFC) is causally linked to value comparison mechanisms [3,6,7],

while the dorsal anterior cingulate (dACC) is associated with learning from feedback with BOLD activity in this region correlated with adapting learning rate [8]. Finally, amygdala (amy) grey matter density increases with experience in reversal learning-like tasks such as ours [9], lesions to the amygdala affect reversal learning [10], and amygdala signals deviation from precise local reward learning [11,12]. Guided by past NHP work, we analysed structural brain data from an independent data set of 125 individuals from the Human Connectome Project data (HCP developmental and young adult data; [13,14]), evenly spread out across our investigated age range. We conducted this study in parallel to study 1. Estimates of individual participants grey matter thickness were extracted from anatomical masks of lateral orbitofrontal cortex and medial orbitofrontal/ventromedial prefrontal cortex [15], dorsal anterior cingulate cortex, amygdala, and anterior insula. Developmental trajectories of all five regions were compared using ANCOVA analysis and showed significant differential GM patterns across age ($F_{4,492}$ = 12.35, $p < 0.001$). Follow-up subanalyses compared lateral orbitofrontal cortex separately with the other four areas. Adults and adolescents were also compared directly in independent samples $t$ tests and Pearson correlational analyses. **(A, B)** Supporting our hypothesis, we showed that grey matter in lateral orbitofrontal cortex was significantly lower in young adults compared to adolescence (independent samples $t$ test, $t_{123}$ = 6.23, $p < 0.001$) and correlated negatively with age (Pearson correlation, R = −0.47, $p < 0.001$; Fig 4B), with link functions suggesting that this relationship was best fit with a quadratic function (Table A in S1 Text). **(C, D)** The GM trajectory of the anterior insula, a region in which BOLD activity correlates with the GRS in macaques [1,9], also showed a significant relationship with age (independent samples $t$ test: $t_{123}$ = −4.34, $p < 0.001$, Pearson correlation, R = −0.39, $p < 0.001$). Follow-up tests suggest that this relationship was best characterised by a quadratic function (Table A in S1 Text). Direct comparison between the GM trajectories of lateral orbitofrontal cortex and the anterior insula revealed no significant differences between the GM trajectory of the two regions ($F_{1,123}$ = 0.16, $p = 0.694$). **(E, F)** The medial orbitofrontal/ventromedial prefrontal cortex also showed continued maturation across the age-range sampled (independent samples $t$ test, $t_{123}$ = 3.19, $p = 0.002$; Pearson correlation, R = −0.21, $p = 0.018$; Fig 4C and 4D) with model fits again characterising this relationship as quadratic (Table A in S1 Text). However, as the ANCOVA results revealed differential developmental trajectories of GM between lateral and medial orbitofrontal cortex, indexed by a significant age × subregion interaction ($F_{1,123}$ = 9.896, $p = 0.002$), which suggested medial maturation was significantly less pronounced than lateral regions. **(G, H)** By contrast, there was no relationship between age and grey matter in the amygdala, a subcortical region heavily connected with lateral orbitofrontal cortex and intrinsically linked to complementary components of local reward learning [10] (independent samples $t$ test, $t_{123}$ = −0.79, $p = 0.43$; Pearson correlation, R = 0.03, $p = 0.716$). Note that this did not improve by using a quadratic instead of a linear link function, see Table A in S1 Text, which replicates past developmental GM studies [16,17]. Direct comparison between the GM trajectories of lateral orbitofrontal cortex and the amygdala showed, as expected, that developmental GM trajectory was significantly more pronounced in lateral prefrontal cortex compared to the amygdala (significant age × region interaction $F_{1,123}$ = 23.37, $p < 0.001$). **(I, J)** Finally, we examined GM trajectory of the anterior cingulate cortex (focusing on the RCZa or more commonly referred to as dorsal ACC). GM in this region did not vary as a function of age (independent samples $t$ test: $t_{123}$ = −0.75, $p = 0.456$, Pearson correlation, R = 0.05, $p = 0.549$). Direct comparison between the GM trajectories of lateral orbitofrontal cortex and the dACC showed, as expected, that developmental GM trajectory was significantly more pronounced in lateral orbitofrontal cortex (significant age × region interaction ($F_{1,123}$ = 24.80, $p < 0.001$). **(K)** Illustration of the between-subjects interaction of the GM maturation (calculated as

mean adolescents minus mean adults) between lateral orbitofrontal cortex, medial orbitofrontal/ventromedial prefrontal cortex, amygdala, dorsal anterior cingulate cortex, and anterior insula. This highlights the significantly stronger maturation of grey matter in lateral orbitofrontal cortex and anterior insula compared to the other networked brain regions. This pattern suggests the lateral orbitofrontal and anterior insula cortex undergo the most extensive changes during adolescence, findings in line with a general pattern of maturation across adolescence [18,19]. This suggests that cognitive functions supported by these regions may also undergo more pronounced changes during development compared to those supported by the other areas in the learning and decision-making network. ("x"s indicate individual participants; plots show mean −/+ SEM; solid line in the right plots indicate a linear fit. Dashed lines represented 95th% confidence intervals. $^*p < 0.05$, $^{**}p < 0.001$). Data for A-K are available in S1 Data (Fig E tab). **Fig F. Study 2: Task design, reward schedule, and lesion overlap in human patients. (A)** Trial timeline: In each testing session, human patients made choices among three novel stimuli (fractal images; left-hand side) via a keyboard button press response. Visual feedback of choice was delivered according to the particular reward schedule (right-hand side). Both possible outcomes are displayed in this example: A green tick was delivered in the case of a positive outcome (top panel) and a red cross during no reward events (bottom panel). **(B)** Reward schedules comprised three options whose reward probabilities ranged between 0.1 and 1 and drifted throughout the session, with each option being competitive at some time during the session (i.e., each option was the best one at least during a short phase of the session). Participants performed this task twice using the same reward schedule but different stimuli. **(C)** Medial (Left) and lateral frontal lobe (right) lesion outlines as based on patients' most recent scan represented on the MNI standard template. Colorbar indicates lesion overlap ($n = 4$ and $n = 4$, respectively). **Fig G. Study 3: Task design, reward schedule, and lesion. (A)** Trial timeline: In each testing session, macaques made choices among three novel stimuli (novel clip art images; left-hand side) via a touch screen before receiving auditory feedback and, according to the particular reward schedule, a sucrose pellet (S) reward or nothing (right-hand side). The chosen stimuli remained onscreen during feedback. Both possible outcomes are displayed in this example: A reward pellet and auditory feedback was delivered in the case of a positive outcome (top panel) and nothing happened during no reward events (bottom panel). **(B)** Animals made choices across three similar reward schedules in which the reward probabilities ranged between 0 and 1 and drifted throughout the session, with each option being competitive at some time during the session (i.e., each option was the best one at least during a short phase of the session). **(C)** Medial (left) and lateral frontal lobe (right) lesion locations represented on an unoperated control, with redness indicating lesion overlap ($n = 4$ and $n = 3$, respectively). **Table A. Summary table of Pearson R values, $R^2$ values, and AIC values for linear and quadratic model fits**. Table shows for key behavioural (percentages or beta weights) and neural (grey matter Jacobean values) results related to age. In all instances, the coefficients of linear or quadratic polynomial fits were compared using a likelihood-ratio test. The best fitting model was indexed by the lowest AIC value. The last row summarises the winning model (linear or quadratic) for the key behavioural analyses and five GM regions of interest; lOFC (lateral orbitofrontal cortex), mOFC (medial orbitofrontal /ventromedial prefrontal cortex), amygdala, dACC (dorsal anterior cingulate), and ains (anterior insula). $^*p < 0.05$.
(DOCX)

**S1 Data. Supplementary data for Figs 1–5 and Figs A-E in S1 Text.**
(XLSX)

## Acknowledgments

The authors would like to acknowledge the early contributions to the study made by Juliette Westbrook, Morwenna Rickards, and Linnet Chan as part of their third year project. Neuro-imaging data were provided by the Human Connectome Project: Young Adult and Development, WU-Minn Consortium funded by the 16 NIH Institutes and Centers that support the NIH Blueprint for Neuroscience Research; and by the McDonnell Center for Systems Neuroscience at Washington University. Human lesion patient data were provided by Lesley Fellows, McGill University. We would also like to thank Miriam Klein-Flügge, Alberto Lazari, Patricia Lockwood, and Matthew Rushworth for their constructive feedback on early versions of the manuscript. We also thank Bolton Chau for allowing us to adapt his analysis scripts. Finally, we are also grateful to all our participants, young and older, human and monkey, for contributing to the study.

## Author Contributions

**Conceptualization:** Marco K. Wittmann, Maximilian Scheuplein, Sophie G. Gibbons, MaryAnn P. Noonan.

**Data curation:** MaryAnn P. Noonan.

**Formal analysis:** Marco K. Wittmann, Sophie G. Gibbons, MaryAnn P. Noonan.

**Funding acquisition:** MaryAnn P. Noonan.

**Investigation:** Maximilian Scheuplein, Sophie G. Gibbons, MaryAnn P. Noonan.

**Methodology:** Marco K. Wittmann, Maximilian Scheuplein, Sophie G. Gibbons, MaryAnn P. Noonan.

**Project administration:** MaryAnn P. Noonan.

**Resources:** MaryAnn P. Noonan.

**Software:** MaryAnn P. Noonan.

**Supervision:** MaryAnn P. Noonan.

**Validation:** MaryAnn P. Noonan.

**Visualization:** Marco K. Wittmann, Sophie G. Gibbons, MaryAnn P. Noonan.

**Writing – original draft:** Marco K. Wittmann, MaryAnn P. Noonan.

**Writing – review & editing:** Marco K. Wittmann, Maximilian Scheuplein, Sophie G. Gibbons, MaryAnn P. Noonan.

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
