## [Editor Report · Decision Letter 0]

8 Apr 2022

Dear Dr Noonan, 

Thank you for submitting your manuscript entitled "Development of local and global reward learning in lateral prefrontal cortex" for consideration as a Research Article by PLOS Biology.

Your manuscript has now been evaluated by the PLOS Biology editorial staff and I am writing to let you know that we would like to send your submission out for external peer review.

Once your full submission is complete, your paper will undergo a series of checks in preparation for peer review. Once your manuscript has passed the checks it will be sent out for review. To provide the metadata for your submission, please Login to Editorial Manager (https://www.editorialmanager.com/pbiology) within two working days, i.e. by Apr 10 2022 11:59PM.

If your manuscript has been previously reviewed at another journal, PLOS Biology is willing to work with those reviews in order to avoid re-starting the process. Submission of the previous reviews is entirely optional and our ability to use them effectively will depend on the willingness of the previous journal to confirm the content of the reports and share the reviewer identities. Please note that we reserve the right to invite additional reviewers if we consider that additional/independent reviewers are needed, although we aim to avoid this as far as possible. In our experience, working with previous reviews does save time. 

If you would like to send previous reviewer reports to us, please email me at kdickson@plos.org to let me know, including the name of the previous journal and the manuscript ID the study was given, as well as attaching a point-by-point response to reviewers that details how you have or plan to address the reviewers' concerns. 

Kind regards,

Kris

Kris Dickson

Neurosciences Senior Editor/Section Manager

PLOS Biology

kdickson@plos.org

---

## [Decision Letter · Decision Letter 1]

31 May 2022

Dear MaryAnn,

Thank you for your considerable patience while your manuscript "Development of local and global reward learning in lateral prefrontal cortex" was peer-reviewed at PLOS Biology. It has now been evaluated by the PLOS Biology editors, an Academic Editor with relevant expertise, and by several independent reviewers. 

In light of the reviews, which you will find at the end of this email, we would like to invite you to revise the work to thoroughly address the reviewers' reports.

As you will see below, the reviewers have provided detailed feedback, including requests for clarifications and suggestions for improving the manuscript that would need to be addressed in your revision. Additionally, an overarching comment from the reviewers is that the integration across the studies could be improved. Methodological challenges inherent in this integration should be acknowledged (while also addressing the question of anatomical specificity of LPFC and MPFC).

Given the extent of revision needed, we cannot make a decision about publication until we have seen the revised manuscript and your response to the reviewers' comments. Your revised manuscript is likely to be sent for further evaluation by all or a subset of the reviewers.

**IMPORTANT - SUBMITTING YOUR REVISION**

*Re-submission Checklist*

*Published Peer Review*

*PLOS Data Policy*

*Blot and Gel Data Policy*

Sincerely,

Kris

Kris Dickson, Ph.D. (she/her)

Neurosciences Senior Editor/Section Manager

PLOS Biology

kdickson@plos.org

REVIEWS:

Reviewer's Responses to Questions

PLOS authors have the option to publish the peer review history of their article (what does this mean?). If published, this will include your full peer review and any attached files.

Reviewer #1: No

Reviewer #2: No

Reviewer #3: No

Reviewer #1: 

To strengthen the theoretical claims and conceptual coherence of this excellent manuscript, I would recommend the following revisions. 

1. Please describe in further detail the relationship(s) between the different pieces of evidence presented in this work, focusing on their integration and the potential methodological limitations this effort faces. 

2. Please include citations to the human lesion literature examining lateral prefrontal cortex lesions and their roles in intelligence, fluid cognition, working memory, and/or other facets of intelligence that bear on the present study (e.g., by Duncan, Grafman, and others). 

3. It would also be valuable to discuss recent work in network neuroscience that examines the role of local and global brain network efficiency in intelligence (e.g., by Barbey, Sporns, and others). 

4. Please describe in greater detail the theoretical claims supported by the present findings and their contribution(s) to the larger literature. 

Reviewer #2: Based on their results of earlier studies with nonhuman primates, the authors have combined three studies in this manuscript: (1) Adolescents and young adults performed a behavioral task online on reward guided choices (as established in earlier studies by the same authors). (2) Data from the Human Connectome Project was used to investigate grey matter maturation in the lateral prefrontal cortex, the medial prefrontal cortex and the amygdala in the same age group as participants in study 1. (3) Lesion patients performed the same task as participants in study 1.

Overall, each individual study seems well conducted and analyzed. I have to admit that I am not able to judge study 2 in detail because I lack hands-on expertise in the methods employed. 

The combination of the findings into one manuscript is laudable and points to an underappreciated role of the lateral prefrontal cortex in (the age-related development) of what the authors call "local and global reward states" in their decisions to switch between choice options that fluctuate in their reward probabilities. 

1. Links between studies: As written above, I truly appreciate the authors' effort to combine these three studies. Nevertheless, my general comment is that the authors should be a bit more specific in developing their hypotheses and their rationale for combining these studies. In my view, the motivation and the inspiration, which in large parts is based on findings from nonhuman primates, becomes clear throughout the manuscript. But it took me quite a bit to grasp this rationale. Let me list some suggestions for rather simple clarifications.

a. Especially the abstract is unclear. The three studies could be listed.

b. The introduction could start with a summary of the findings from nonhuman primates.

c. The words "local and global reward states" are initially quite unclear to readers not familiar with the task. For example, it should be mentioned early on that these words refer to temporal (i.e., trial-by-trial) "adjacency" and not to spatial proximity. Also, "state" does neither refer to "neural states" nor to states as in a Markov Decision Process (as far as I understand the wording used here).

d. It is unclear whether the three studies were conducted in parallel or serially.

e. Related to the previous point: It is unclear whether the lesion patients were a "convenience sample" or whether they were selectively recruited for this study (e.g., it could be that they were specifically recruited after the completion of study 2).

f. The same point applies to the links between studies 1 and 2. Were the datasets from the Human Connectome Project selected after the behavioral results from study 1 were known?

2. Analyses and modelling of the behavioral data: The authors conducted a fine-grained series of analyses. In particular, they use three sets of regressors (CxR-history, C-history, and GRS). They also test the choice frequency of the "highest value-option (as defined by value estimates from a Rescorla-Wagner based reinforcement learning model, see Methods)." 

a. I may have missed this, but I could not find more details on the Rescorla-Wagner model in the methods. 

b. More importantly, I am wondering if variants of the Rescorla-Wagner model could be directly fitted to the behavioral data. Such models could provide a complementary but more specific - and at the same time more general - outlook on the data than the set of regressors. E.g., the choice of three preceding trials for averaging seems a bit arbitrary; fitting a learning rate as a free parameter could be more generic. Have the authors tested some of these models in the past? Such models would allow to correlate fitted parameters with age.

3. Specificity of the task: I am wondering how much task performance across participants differing in age or lesion site depends on the task itself. In particular, on the volatility or the rate of how fast the reward probabilities change (traces in Figure 1B). 

a. How did the authors choose the changes in reward probabilities? Did they vary these in earlier studies? 

b. It would be interesting to speculate (or even to test) if differences between participants would disappear with lower (or higher) volatility. 

c. This may be outside the scope of this manuscript: Simulations with the help of Rescorla-Wagner models (as suggested in comment 2) could help to see in which task situations particular participants would perform the best/worst. This could give further indications about the role of the lateral prefrontal cortex. Such simulations could also provide clearer links between the Rescorla-Wagner models and the set of regressors.

4. Anatomical masks: I am wondering about grey matter changes in slightly more dorsal parts of the medial prefrontal cortex. As far as I know, these more dorsal regions also change quite substantially during adolescence - and are involved in decision-making.

Reviewer #3: This study combines behavioral, anatomical, and lesion data from three separate studies to argue that distinct local and global components of reward learning develop over adolescence and into young adulthood, and that these behavioral changes are linked to the maturation of lateral prefrontal cortex (LPFC). The authors first use an established probabilistic reward learning paradigm and modeling methods in a large online behavioral study to show that learning of both local reward contingencies, often referred to as credit assignment, and of the global reward state (i.e., average recent reward history regardless of choice) strengthens across development. They then test an a priori hypothesis based on prior work in humans and non-human primates that these behavioral effects depend specifically on LPFC through separate analyses of gray matter maturation across development and behavior on the same task in human lesion patients. Specifically, they use a subsample of the Human Connectome Project (HCP) dataset to show that LPFC gray matter (GM) volume matures (decreases) throughout the same age range as the large-scale behavioral study (11-35), that that this effect is larger than maturation in medial prefrontal cortex (MPFC), and that an amygdala control region does not change over the same age range. Lastly, they show that patients with lesions in LPFC are impaired in both local and global reward learning compared to those with MPFC lesions. By combining these multiple lines of evidence, the authors argue that LPFC is "a strong potential candidate to underlie the behavioral differences in local and global reward learning" seen across this period in development.

The manuscript is well written and addresses the important issue of the development of reward learning and decision making, which is of interest to a wide audience of neuroscientists, psychologists, and developmental researchers. The authors apply a relatively novel framework based on computational modeling of reward learning that separates local learning about specific choices from global learning about the overall reward environment. This distinction has potential utility for understanding the role of risk and uncertainty in adolescent decision making, which justifies exploration of these effects across development. Another strength of the paper is that this framework can be interpreted through the lens of optimal or adaptive foraging behavior, and the authors do a good job of interpreting these two components of learning and their behavioral results from the perspectives of behavioral ecology and evolution, which also helps broadens the potential audience.

The most important findings in the study are from the behavioral study, which shows nicely that both local and global reward learning improve from adolescence into young adulthood. These conclusions are robust thanks to multiple converging results, including both group comparisons between adolescents and young adults and linear correlations across both age ranges. Another methodological strength is the use of model comparisons to determine whether linear and quadratic functions explain behavioral and neural changes across time, though it is unclear how to interpret the relationships between linear changes in local and global reward learning and quadratic changes in overall task performance and gray matter maturation. The authors also include control analyses showing age is unrelated to the choice history regressor in the credit assignment GLM, the inverse temperature parameter from standard reinforcement learning models ("decision-making noise"), and the "bias by irrelevant alternative" effect previously linked to MPFC (Noonan et al., 2017).

Additionally, the correlation between these two components of learning across participants is interesting and suggests their development may be linked, but this link could be mediated by other variables besides LPFC maturation. In particular, there is a significant increase across development in the learning rate parameter from the reinforcement learning model, so it would be interesting to know whether this general parameter mediates either or both of the local or global reward learning effects. If it mediates both effects, this could explain the correlation between the two metrics, and one could ask why researchers should study local and global learning parameters instead of the single, more general (and parsimonious) learning rate. On the other hand, showing that the effect on general learning rate explains only one or neither would demonstrate the additional predictive power of local contingency learning and global reward state variables and emphasize their importance in studies of reward learning and value-based decision-making. Relatedly, the authors could improve the interpretation of these constructs and establish their utility by showing that improvements in local and/or global reward learning predict success on the task (e.g., total rewards, % best choice, etc.), especially since these performance metrics also improve across development (Fig. 2). 

Perhaps the most novel and interesting finding reported in the manuscript is the negative effect of global reward state (GRS) on win-stay lose-switch (WSLS) behavior, which increases across development and was confirmed via both credit assignment GLM and ANOVA-based analyses. This is remarkable given that a positive effect in the opposite direction was previously reported in non-human primates (NHPs; Whittmann et al., 2020). The negative relationship found here is optimal in the current task and, as argued at the end of Study 1 in the Results section, is considered adaptive in the context of behavioral ecology, though this seems to slightly contradict arguments presented in the previous paper by the same first author that the positive effect observed in NHPs was potentially adaptive in natural environments with serial dependencies. Regardless, this species difference is a striking finding that deserves to be highlighted, and it will be interesting to see if it replicates across different task structures and whether this pattern is unique to humans. That said, the discussion of how the conceptual framework and empirical findings help us understand decision making under risk and uncertainty during adolescence is not particularly compelling, and given the emphasis on this topic in the abstract, these arguments could be better developed to integrate the behavioral ecology and developmental perspectives.

In contrast, the evidence linking local and global reward learning to LPFC in this manuscript is flawed and provides little confidence that developmental changes in local and global reward learning are related to maturation of LPFC. The biggest issues are related to the weak logic behind the GM volume analyses. Showing that LPFC changes in GM volume in a separate cross-sectional sample from the HCP merely confirms what has been previously published, mainly that maturation of PFC as a whole (e.g., as shown in Gogtay et al., 2004 or Raznahan et al., 2010) also applies to specific LPFC subregions 47/12o and 14. More importantly, showing that two variables (behavior and LPFC GM) change across the same age range in two non-overlapping samples does not provide evidence that those effects are related. Similarly, showing that LPFC is maturing to a greater degree than MPFC over the same period is not sufficient to claim one is more closely related to a behavioral change. Without stronger evidence (e.g., by measuring these trends in the same sample), the claims based on the HCP data are weakly supported.

A second major concern is the use of the broad terms "LPFC" and "MPFC", which are problematic for several reasons. First and most generally, it is important to be as precise as possible with neuroanatomical terminology. As noted several times in the manuscript, two recent NHP papers published by the first author used similar tasks and analyses in two recent NHP papers to show local contingency learning is specific and causally related to area 47/12o in the ventrolateral PFC, and that this effect is dissociable from GRS signals localized to the nearby but distinct anterior insular cortex (AIC) (Whittmann et al., 2020; Folloni et al., 2021). Therefore, it's unclear why the manuscript uses the general term LPFC when the strong evidence for their a priori hypotheses is much more specific. In particular, this deceives the reader into thinking both functions rely on the same region, when in fact these two different cognitive processes have been carefully dissociated in prior studies. For example, the lack of any ROI covering the AIC region linked to GRS in their GM analyses leads the reader to assume that both functions have been localized to the 47/12o ROI. Moreover, the AIC GRS region is not even a subregion of PFC, meaning the broad LPFC label is not even an accurate superset of the two subregions. This framing is therefore oversimplified to the point that it is a disservice to the precision of the prior work and muddies the waters for the field in general, especially for readers not familiar with these prior studies, these local and global components of reward learning, or the relevant neuroanatomy. Similarly, the authors seem to mean a specific ventromedial PFC region when they refer to MPFC (which covers a much larger range of regions and functions), and if that's the case, it would improve the clarity of this manuscript and the literature as a whole to use the more precise term vmPFC instead of MPFC.

I suspect the authors made this choice because they wanted to include the human lesion data, which are inherently limited in spatial resolution. These results are valuable and provide causal evidence that local and global reward learning are more specific to lateral than medial regions, which adds an important brain-behavior link that was missing from the gray matter findings. While the diffuse nature of the lesions likely renders this analysis unable to separate the two subregions and dissociate the two cognitive functions, this is reasonable and a standard challenge for human lesion data. The authors could simply acknowledge that these two processes rely on different subregions that are both covered by the LPFC lesions, which would maintain the distinction in the literature and improve the interpretability of the current study. In that case, showing a difference in both behaviors between groups with medial and lateral lesions is still a strong finding. However, another issue is that this difference is hard to interpret without behavioral data from age-matched controls, especially since these lesion patients are older adults that may behave differently than the adolescents and young adults. At the very least, the age of the lesion patients should be included in the Methods section. Additionally, it is generally important to be cautious when interpreting human lesion data because they typically cover multiple subregions as well as neighboring white matter tracts, particularly since these issues and related differences between lesion methods and locations across rodent, NHP, and human data have generated conflicting findings and confusion in the literature (see Sallet et al., 2020 and related reversal learning literature).

In summary, the manuscript has an interesting and novel set of findings about how local and global components of reward learning change across development, and the lesion data provide evidence that these functions are causally related to lateral regions and not MPFC in humans. However, this brain-behavior link is based on older adults, which leaves the behavioral study as the only convincing developmental findings. While the neuroanatomical language can potentially be improved, the gray matter analyses meant to address brain development are flawed enough that they detract more than they add to the paper. In my opinion, removing the manuscript would be simplified by replacing that analysis with a few strong references. Unfortunately, this leaves the claims linking behavioral changes across development to neural maturation poorly supported. Thus, the manuscript as currently written contains important and novel contributions to human reward neuroscience based on the interesting behavioral effects and lesion data, but with only weak evidence linking these behavioral changes to brain development, it offers fairly limited contributions related to adolescence that would appeal to the wider audience of developmental researchers. Therefore, in my opinion, further evidence is needed to justify framing the manuscript around development, or lacking such evidence, it should be rewritten as a basic reward neuroscience paper. 

Minor Grammatical points:

Introduction paragraph 2, sentence 3: "particularly in LPFC" should likely be "particularly those related to LPFC" since cognitive ablities cannot be "in" LPFC

Supp. Table 1 legend: "a likelihood ratio tests" grammar

Discussion: "behaviroual " misspelled

Methods: missing verb: "Young adults were additionally via local university online advertisement and via email lists. "; add "to" to "These precise regions have specifically and repeatedly been linked learning and decision-making mechanisms in the adult population "

Upon looking over the other reviewer comments, I have a few additional thoughts:

* The behavioral analysis approach, including the conceptual framework and findings, seem likely to be useful in the debate over decision making in adolescence, so if the concerns around that study and its interpretation are addressed, this is already a strong behavioral paper. Adding the lesion data to incorporate the "biology" parts makes it a very nice reward neuroscience paper. The gray matter analyses are not convincing to me, but I'm not an expert in that field so perhaps that result is more novel than it seems. I think if the authors did a better job of integrating the results from the three studies, the gray matter result could be positioned as tenuous evidence that the lesion result is in a region that develops in this age range.

* I do wish there was more discussion of how these findings integrate with the developmental literature, but I also recognize that fully developing arguments around how those issues can be viewed through their conceptual framework is a tall task that would be tough to fit in a reasonable Discussion section and may better fit as a review/position/opinion paper.

---

## [Decision Letter · Decision Letter 2]

9 Jan 2023

Dear Dr Noonan,

Thank you for your patience while we considered your revised manuscript "Development of local and global reward learning in lateral frontal cortex" for publication as a Research Article at PLOS Biology. This revised version of your manuscript has been evaluated by the PLOS Biology editors, the Academic Editor and two of the original reviewers.

Based on the reviews and our discussion with the Academic Editor, we are likely to accept this manuscript for publication, provided you satisfactorily address the minor suggestions raised by Reviewer 3, consider the following editorial request, and address the data and other policy-related requests listed at the bottom of this email. Please note that all of the data/policy requests must be addressed in full for us to move forward with this work.

***Editorial request:

Please consider a minor title change to more clearly convey your interesting findings to our broad neuroscience audience:

Local and global reward learning show differential development in the lateral frontal cortex during human adolescence

Please also take this last chance to review your reference list to ensure that it is complete and correct. If you have cited papers that have been retracted, please include the rationale for doing so in the manuscript text, or remove these references and replace them with relevant current references. Any changes to the reference list should be mentioned in the cover letter that accompanies your revised manuscript.

We expect to receive your revised manuscript within two weeks. 

*Published Peer Review History*

*Press*

Sincerely,

Kris

Kris Dickson, Ph.D., (she/her)

Neurosciences Senior Editor/Section Manager,

kdickson@plos.org,

PLOS Biology

DATA POLICY:

You may be aware of the PLOS Data Policy, which requires that all data be made available without restriction: http://journals.plos.org/plosbiology/s/data-availability. For more information, please also see this editorial: http://dx.doi.org/10.1371/journal.pbio.1001797. Note that we do not require all raw data. Rather, we ask that all individual quantitative observations that underlie the data summarized in the figures and results of your paper be made available.

We note that you have indicated that the developmental dataset with be anonymized and available "on request" and that permissions are not in place to share the human or macaque lesion data. Before moving forward, we will need you to either provide anonymized summary data for all of the figures in your manuscript, or have you provide clear justification for why you are unable to do so (e.g. if the anonymized data is 3rd party or allows individual identification, or if there is a problem of patient privacy, this will need to be stated in the manuscript as the reason for the limitation in providing the summary data).

Summary data can be supplied in one of two formats:

1) Supplementary files (e.g., excel). Please ensure that all data files are uploaded as 'Supporting Information'.

Regardless of the method selected, please ensure the data are invariably referred to (in the manuscript, figure legends, and the Description field when uploading your files) using the following format verbatim: S1 Data, S2 Data, etc. Multiple panels of a single or even several figures can be included as multiple sheets in one excel file that is saved using exactly the following convention: S1_Data.xlsx (using an underscore)

**Please ensure that you provide the individual numerical values that underlie the summary data displayed in the following figure panels as they are essential for readers to assess your analysis and to reproduce it:

Fig 1B,C; Fig2A-D; Fig3B-G; Fig4, Fig5A,B

Supplemental FigS1A,B; FigS2A-F; FigS3A,B; FigS4, FigS5A-K, FigS6B; FigS7B

**Please also ensure that figure legends in your manuscript include information on where the underlying data can be found, and ensure your supplemental data file/s has a legend.

**Please ensure that your Data Statement in the submission system accurately describes where your data can be found.

DATA NOT SHOWN?

- Please note that per journal policy, we do not allow the mention of "data not shown", "personal communication", "manuscript in preparation" or other references to data that is not publicly available or contained within this manuscript. Please check your submission carefully and either remove mention of any such data or provide figures presenting the results and the data underlying the figure(s).

Reviewer remarks:

Reviewer's Responses to Questions

Do you want your identity to be public for this peer review?

Reviewer #2: Yes: Christoph Korn

Reviewer #3: Yes: Colin W Hoy

Reviewer #2: All my comments have been addressed.

Reviewer #3: Overall, this is a strong manuscript that makes combined behavioral and neural contributions to the field of reward learning and development. I would like to thank the authors for their efforts to revise and improve the manuscript, which have addressed all of my major concerns. This involved major reorganization of the manuscript, including revisions to the structure of their arguments in the introduction and discussion, as well as new data, analyses, and results. I believe that the additional analyses clarifying the relationship between local and global reward learning and more broadly used reinforcement learning parameters such as learning rate will help contextualize these concepts in the wider field, and the new language helping interpret these findings from the perspectives of behavioral ecology and development ensure those advances are accessible to audience beyond cognitive neuroscience. The HCP GM volume data is appropriately still included as a supplement, and breaking that analysis down according to the specific subregions associated with local and global reward learning respectively improves their contribution. More generally, the authors are careful to use specific neuroanatomical terms when possible but general terms when necessary (i.e., referring to lesion locations). Most importantly, the new non-human primate lesion and behavioral data and analyses reinforce and clarify the striking species differences in the effect of GRS on decision-making, strengthening a major contribution to the field. See below for minor comments and suggestions, but I would recommend this manuscript for publication and would be comfortable with the authors making these minor corrections without further consultation of the reviewers.

Minor issues:

In the introduction, the authors refer to the insula as a subcortical region when it is in fact a cortical region (sentence 2).

In the results, Fig. 1a shows two examples of positive feedback instead of a positive and negative example. A similar issue appears in Sup. Fig. 7a.

For Supplemental Fig. 4, it's a little unclear how the results were aggregated across the analyses using different history lengths to make the correlation plot, but perhaps I missed that somewhere. If not, that language in the methods or caption could be further clarified.

Optional suggestions:

The language emphasizing the differences between the effect of lateral frontal lesions making the effect of GRS on local reward learning weaker vs. stronger for humans vs. NHPs respectively is appropriate, but there's also a point of potential reconciliation: the overall direction of the effect (more positive GRS effect) is consistent across species. Obviously, the baseline difference changes the interpretation, but is it possible LPFC is playing a general role in contextualizing GRS to avoid the "spread of reward"? Perhaps this effect is already strong enough in humans to create a negative overall effect, but this function degrades in both species after LPFC lesions. Granted this perspective is speculative, but I just wanted to bring it to the authors attention as a potential discussion point.

The authors might also consider adding a citation diversity statement (see Zurn, Bassett, & Rust 2020 Trends in Cog. Sci. for rationale, as well as references to original work and a template and tools for implementation) to help combat systemic inequalities in academia.

Minor grammatical changes:

In general, there were more typos and grammatical errors in this version of the manuscript (as often happens in revisions). I would recommend a thorough pass on the language, but here are a few I happened to write down while reading:

Abstract:

* "anterior insular cortex" when insula is used as an adjective

Introduction: 

* Missing word in "more rapid maturation of subcortical ___"

* Generally watch for commas needed between sentences, e.g., after refs [35,36]

Results:

* "The second reason was to examine, for the first time, whether changes in global reward learning were also apparent after lateral frontal lobe LESIONS in macaques, like in our human lesion data. 

Discussion:

* "(Fig.4,5). This suggests that …"

* ", in increased the effect in macaques."

Supplement:

* S4 caption: "Consequently, in the analysis, variance associated with one learning mechanism is "

---

## [Editor Report · Decision Letter 3]

20 Jan 2023

Dear Dr Noonan,

Thank you for the submission of your revised Research Article "Local and global reward learning in the lateral frontal cortex show differential development during human adolescence" for publication in PLOS Biology. On behalf of my colleagues and the Academic Editor, Robert Whelan, I am pleased to say that we can in principle accept your manuscript for publication, provided you address any remaining formatting and reporting issues. These will be detailed in an email you should receive within 2-3 business days from our colleagues in the journal operations team; no action is required from you until then. Please note that we will not be able to formally accept your manuscript and schedule it for publication until you have completed any requested changes. When you log in to make these additional changes, please also ensure that you update your Data Availability statement on the Details section of your submission. The answers on this page are pulled into the final version of your work, so it needs to accurately reflect the changes you've made. Please update this to say:

Data Availability: Summary data are available as a supplemental excel spreadsheet (S1_data.xlsx) in which each tab

summarises the data to reproduce each figure. Multiple figure panels are included on each tab.

Please also take a minute to log into Editorial Manager at http://www.editorialmanager.com/pbiology/, click the "Update My Information" link at the top of the page, and update your user information to ensure an efficient production process.

PRESS

Sincerely, 

Kris

Kris Dickson, Ph.D., (she/her)

Neurosciences Senior Editor/Section Manager

PLOS Biology

kdickson@plos.org